# Fast and Flexible Temporal Point Processes with Triangular Maps

**Oleksandr Shchur, Nicholas Gao, Marin Biloš, Stephan Günnemann**
Technical University of Munich, Germany
{shchur,gaoni,bilos,guennemann}@in.tum.de

## Abstract

Temporal point process (TPP) models combined with recurrent neural networks provide a powerful framework for modeling continuous-time event data. While such models are flexible, they are inherently sequential and therefore cannot benefit from the parallelism of modern hardware. By exploiting the recent developments in the field of normalizing flows, we design TriTPP— a new class of non-recurrent TPP models, where both sampling and likelihood computation can be done in parallel. TriTPP matches the flexibility of RNN-based methods but permits orders of magnitude faster sampling. This enables us to use the new model for variational inference in continuous-time discrete-state systems. We demonstrate the advantages of the proposed framework on synthetic and real-world datasets.

## 1 Introduction

Temporal data lies at the heart of many high-impact machine learning applications. Electronic health records, financial transaction ledgers and server logs contain valuable information. A common challenge encountered in all these settings is that both the number of events and their times are variable. The framework of temporal point processes (TPP) allows us to naturally handle data that consists of variable-number events in continuous time. Du et al. [1] have shown that the flexibility of TPPs can be improved by combining them with recurrent neural networks (RNN). While such models are expressive and can achieve good results in various prediction tasks, they are poorly suited for sampling: sequential dependencies preclude parallelization. We show that it's possible to overcome the above limitation and design flexible TPP models without relying on RNNs. For this, we use the framework of triangular maps [2] and recent developments in the field of normalizing flows [3].

Our main contributions are: **(1)** We propose a new parametrization for several classic TPPs. This enables efficient parallel likelihood computation and sampling, which was impossible with existing parametrizations. **(2)** We propose TriTPP— a new class of non-recurrent TPPs. TriTPP matches the flexibility of RNN-based methods, while allowing orders of magnitude faster sampling. **(3)** We derive a differentiable relaxation for non-differentiable sampling-based TPP losses. This allows us to design a new variational inference scheme for Markov jump processes.

## 2 Background

**Temporal point processes (TPP)** [4] are stochastic processes that model the distribution of discrete events on some continuous time interval $[0, T]$. A realization of a TPP is a *variable-length* sequence of strictly increasing arrival times $\boldsymbol{t} = (t_1, \ldots, t_N), t_i \in [0, T]$. We make the standard assumption and focus our discussion on regular finite TPPs [4]. One way to specify such a TPP is by using the (strictly positive) conditional intensity function $\lambda^*(t) := \lambda(t|\mathcal{H}_t)$ that defines the rate of arrival of new events

---

Code and datasets are available under www.daml.in.tum.de/triangular-tpp

given the history $\mathcal{H}_t = \{t_j : t_j < t\}$. The $*$ symbol reminds us of the dependence on the history [5]. Equivalently, we can consider the *cumulative* conditional intensity $\Lambda^*(t) := \Lambda(t|\mathcal{H}_t) = \int_0^t \lambda^*(u)du$, also known as the compensator.[1] We can compute the likelihood of a realization $\boldsymbol{t}$ on $[0,T]$ as

$$p(\boldsymbol{t}) = \left(\prod_{i=1}^N \lambda^*(t_i)\right) \exp\left(-\int_0^T \lambda^*(u)du\right) = \left(\prod_{i=1}^N \frac{\partial}{\partial t_i}\Lambda^*(t_i)\right)\exp\left(-\Lambda^*(T)\right) \qquad (1)$$

For example, we can use a TPP to model the online activity of a user in a 24-hour interval. In this case, each realization $\boldsymbol{t}$ could correspond to the timestamps of the posts by the user on a specific day.

**Triangular maps** [2] provide a framework that connects autoregressive models, normalizing flows and density estimation. Bogachev et al. [6] have shown that any density $p(\boldsymbol{x})$ on $\mathbb{R}^N$ can be equivalently represented by another density $\tilde{p}(\boldsymbol{z})$ on $\mathbb{R}^N$ and an increasing differentiable triangular map $\boldsymbol{F} = (f_1,\ldots,f_N) : \mathbb{R}^N \to \mathbb{R}^N$ that pushes forward $p$ into $\tilde{p}$.[2] A map $\boldsymbol{F}$ is called triangular if each component function $f_i$ depends only on $(x_1,\ldots,x_i)$ and is an increasing function of $x_i$. Intuitively, we can think of $\boldsymbol{F}$ as converting a random variable $\boldsymbol{x} \sim p$ into a random variable $\boldsymbol{z} := \boldsymbol{F}(\boldsymbol{x})$ with a density $\tilde{p}$. We can compute the density $p(\boldsymbol{x})$ using the change of variables formula

$$p(\boldsymbol{x}) = |\det J_{\boldsymbol{F}}(\boldsymbol{x})|\,\tilde{p}\left(\boldsymbol{F}(\boldsymbol{x})\right) = \left(\prod_{i=1}^N \frac{\partial}{\partial x_i} f_i(x_1,\ldots,x_i)\right)\tilde{p}\left(\boldsymbol{F}(\boldsymbol{x})\right) \qquad (2)$$

where $\det J_{\boldsymbol{F}}(\boldsymbol{x})$ is the Jacobian determinant of $\boldsymbol{F}$ at $\boldsymbol{x}$. Here, we used the fact that $J_{\boldsymbol{F}}(\boldsymbol{x})$ is a positive-definite lower-triangular matrix. To specify a complex density $p(\boldsymbol{x})$, we can pick some simple density $\tilde{p}(\boldsymbol{z})$ and learn the triangular map $\boldsymbol{F}$ that pushes $p$ into $\tilde{p}$. It's important that $\boldsymbol{F}$ and its Jacobian determinant can be evaluated efficiently if we are learning $p(\boldsymbol{x})$ via maximum likelihood. We can sample from $p(\boldsymbol{x})$ by applying the inverse map $\boldsymbol{F}^{-1}$ to the samples drawn from $\tilde{p}(\boldsymbol{z})$. Note that $\boldsymbol{F}^{-1} : \mathbb{R}^N \to \mathbb{R}^N$ is also an increasing differentiable triangular map. Fast computation of $\boldsymbol{F}^{-1}$ is important when learning $p(\boldsymbol{x})$ via sampling-based losses (e.g., in variational inference).

## 3 Defining temporal point processes using triangular maps

We can notice the similarity between the right-hand sides of Equations 1 and 2, which seems to suggest some connection between TPPs and triangular maps. Indeed, it turns out that triangular maps can also be used to specify densities of point processes. Let $\boldsymbol{t} = (t_1,\ldots,t_N)$ be a realization of a TPP on $[0,T]$ with compensator $\Lambda^*$ (i.e. with density $p(\boldsymbol{t})$). The random time change theorem states that in this case $\boldsymbol{z} = (\Lambda^*(t_1),\ldots,\Lambda^*(t_N))$ is a realization of a homogeneous Poisson process (HPP) with unit rate on the interval $[0,\Lambda^*(T)]$ [4, Theorem 7.4.I][5, Proposition 4.1] (Figure 1).

The transformation $\boldsymbol{F} = (f_1,\ldots,f_N) : \boldsymbol{t} \mapsto \boldsymbol{z}$ is an increasing triangular map. Each component function $f_i(\boldsymbol{t}) = \Lambda(t_i|t_1,\ldots,t_{i-1})$ only depends on $(t_1,\ldots,t_i)$ and is increasing in $t_i$ since $\frac{\partial}{\partial t_i}\Lambda^*(t_i) = \lambda^*(t_i) > 0$. The number $N$ of the component functions $f_i$ depends on the length of the specific realization $\boldsymbol{t}$. Notice that the term $\prod_{i=1}^N \frac{\partial}{\partial t_i}\Lambda^*(t_i)$ in Equation 1 corresponds to the Jacobian determinant of $\boldsymbol{F}$. Similarly, the second term, $\tilde{p}(\boldsymbol{z}) = \tilde{p}(\boldsymbol{F}(\boldsymbol{t})) = \exp(-\Lambda^*(T))$, corresponds to the density of a HPP with unit rate on $[0,\Lambda^*(T)]$ for any realization $\boldsymbol{z}$. This demonstrates that all TPP densities (Equation 1) correspond to increasing triangular maps (Equation 2). As for the converse of this statement, every increasing triangular map that is bijective on the space of increasing sequences defines a valid TPP (see Appendix C.3).

Our main idea is to define TPP densities $p(\boldsymbol{t})$ by directly specifying the respective maps $\boldsymbol{F}$. In Section 3.1, we show how maps that satisfy certain properties allow us to efficiently compute density and generate samples. We demonstrate this by designing a new parametrization for several established models in Section 3.2. Finally, we propose a new class of fast and flexible TPPs in Section 3.3.

### 3.1 Requirements for efficient TPP models

**Density evaluation.** The time complexity of computing the density $p(\boldsymbol{t})$ for various TPP models can be understood by analyzing the respective map $\boldsymbol{F}$. For a general triangular map $\boldsymbol{F} : \mathbb{R}^N \to \mathbb{R}^N$, computing $\boldsymbol{F}(\boldsymbol{t})$ takes $\mathcal{O}(N^2)$ operations. For example, this holds for Hawkes processes with

arbitrary kernels [7]. If the compensator $\Lambda^*$ has Markov property, the complexity of evaluating $\boldsymbol{F}$ can be reduced to $\mathcal{O}(N)$ *sequential* operations. This class of models includes Hawkes processes with exponential kernels [8, 9] and RNN-based autoregressive TPPs [1, 10, 11]. Unfortunately, such models do not benefit from the parallelism of modern hardware. Defining an efficient TPP model will require specifying a forward map $\boldsymbol{F}$ that can be computed in $\mathcal{O}(N)$ *parallel* operations.

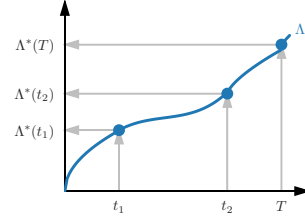

Figure 1: Triangular map $\boldsymbol{F}(\boldsymbol{t}) = (\Lambda^*(t_1), ..., \Lambda^*(t_N))$ is used for computing $p(\boldsymbol{t})$.

**Sampling.** As a converse of the random time change theorem, we can sample from a TPP density $p(\boldsymbol{t})$ by first drawing $\boldsymbol{z}$ from an HPP on $[0, \Lambda^*(T)]$ and applying the inverse map, $\boldsymbol{t} = \boldsymbol{F}^{-1}(\boldsymbol{z})$ [4]. There are, however, several caveats to this method. Not all parametrizations of $\boldsymbol{F}$ allow computing $\boldsymbol{F}^{-1}(\boldsymbol{z})$ in closed form. Even if $\boldsymbol{F}^{-1}$ is available, its evaluation for most models is again sequential [1, 9]. Lastly, the number of points $N$ that will be generated (and thus $\Lambda^*(T)$ for HPP) is not known in advance. Therefore, existing methods typically resort to generating the samples one by one [5, Algorithm 4.1]. We show that it's possible to do better than this. If the inverse map $\boldsymbol{F}^{-1}$ can be applied in parallel, we can produce large batches of samples $t_i$, and then discard the points $t_i > T$ (Figure 2). Even though this method may produce samples that are later discarded, it is much more efficient than sequential generation on GPUs (Section 6.1).

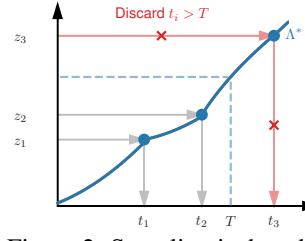

Figure 2: Sampling is done by applying $\boldsymbol{F}^{-1}$ to a sample $\boldsymbol{z}$ from a HPP with unit rate.

To summarize, defining a TPP efficient for both density computation and sampling requires specifying a triangular map $\boldsymbol{F}$, such that both $\boldsymbol{F}$ and its inverse $\boldsymbol{F}^{-1}$ can be evaluated analytically in $\mathcal{O}(N)$ *parallel* operations. We will now show that maps corresponding to several classic TPP models can be defined to satisfy these criteria.

### 3.2 Fast temporal point process models

**Inhomogeneous Poisson process (IPP)** [4] is a TPP whose conditional intensity doesn't depend on the history, $\Lambda(t|\mathcal{H}_t) = \Lambda(t)$. The corresponding map is $\boldsymbol{F} = \boldsymbol{\Lambda}$, where $\boldsymbol{\Lambda}$ simply applies the function $\Lambda : [0, T] \to \mathbb{R}_+$ elementwise to the sequence $(t_1, ..., t_N)$.

**Renewal process (RP)** [12] is a TPP where each inter-event time $t_i - t_{i-1}$ is sampled i.i.d. from the same distribution with the cumulative hazard function $\Phi : \mathbb{R}_+ \to \mathbb{R}_+$. The compensator of an RP is $\Lambda(t|\mathcal{H}_t) = \Phi(t - t_i) + \sum_{j=1}^{i} \Phi(t_j - t_{j-1})$, where $t_i$ is the last event before $t$. The triangular map of an RP can be represented as a composition $\boldsymbol{F} = \boldsymbol{C} \circ \boldsymbol{\Phi} \circ \boldsymbol{D}$, where $\boldsymbol{D} \in \mathbb{R}^{N \times N}$ is the pairwise difference matrix, $\boldsymbol{C} \equiv \boldsymbol{D}^{-1} \in \mathbb{R}^{N \times N}$ is the cumulative sum matrix, and $\boldsymbol{\Phi}$ applies $\Phi$ elementwise.

**Modulated renewal process (MRP)** [13] generalizes both inhomogeneous Poisson and renewal processes. The cumulative intensity is $\Lambda(t|\mathcal{H}_t) = \Phi(\Lambda(t) - \Lambda(t_i)) + \sum_{j=1}^{i} \Phi(\Lambda(t_j) - \Lambda(t_{j-1}))$. Again, we can represent the triangular map of an MRP as a composition, $\boldsymbol{F} = \boldsymbol{C} \circ \boldsymbol{\Phi} \circ \boldsymbol{D} \circ \boldsymbol{\Lambda}$.

All three above models permit fast density evaluation and sampling. Since $\boldsymbol{\Phi}$ and $\boldsymbol{\Lambda}$ (as well as their inverses $\boldsymbol{\Phi}^{-1}$ and $\boldsymbol{\Lambda}^{-1}$) are elementwise transformations, they can obviously be applied in $\mathcal{O}(N)$ parallel operations. Same holds for multiplication by the matrix $\boldsymbol{D}$, as it is bidiagonal. Finally, the cumulative sum defined by $\boldsymbol{C}$ can also be computed in parallel in $\mathcal{O}(N)$ [14]. Therefore, by reformulating IPP, RP and MRP using triangular maps, we can satisfy our efficiency requirements.

**Parametrization for $\Phi$ and $\Lambda$** must satisfy several conditions. First, to define a valid TPP, $\Phi$ and $\Lambda$ have to be positive, strictly increasing and differentiable. Next, both functions, their derivatives (for density computation) and inverses (for sampling) must be computable in closed form to meet the efficiency requirements. Lastly, we want both functions to be highly flexible. Constructing such functions is not trivial. While IPP, RP and MRP are established models, none of their existing parametrizations satisfy all the above conditions simultaneously. Luckily, the same properties are necessary when designing normalizing flows [15]. Recently, Durkan et al. [3] used rational quadratic splines (RQS) to define functions that satisfy our requirements. We propose to use RQS to define $\Phi$ and $\Lambda$ for (M)RP and IPP. This parametrization is flexible, while also allowing efficient density evaluation and sampling — something that existing approaches are unable to provide (see Section 5).

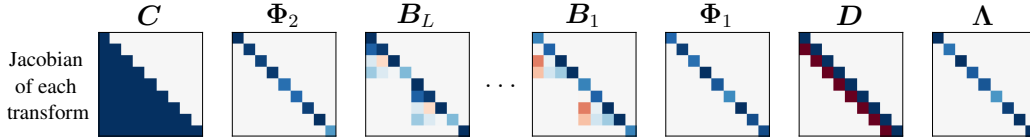

Figure 3: TriTPP defines an expressive map $\boldsymbol{F}$ as a composition of easy-to-invert transformations.

### 3.3 Defining more flexible triangular maps

Even though the splines can make the functions $\Phi$ and $\Lambda$ arbitrarily flexible, the overall expressiveness of MRP is still limited. Its conditional intensity $\lambda^*(t)$ depends only on the global time and the time since the last event. This means, MRP cannot capture, e.g., self-exciting [7] or self-correcting [16] behavior. We will now construct a model that is more flexible without sacrificing the efficiency.

The efficiency of the MRP stems from the fact that the respective triangular map $\boldsymbol{F}$ is defined as a composition of easy-to-invert transformations. More specifically, we are combining *learnable* element-wise nonlinear transformations $\boldsymbol{\Phi}$ and $\boldsymbol{\Lambda}$ with *fixed* lower-triangular matrices $\boldsymbol{D}$ and $\boldsymbol{C}$. We can make the map $\boldsymbol{F}$ more expressive by adding *learnable* lower-triangular matrices into the composition. Using full $N \times N$ lower triangular matrices would be inefficient (multiplication and inversion are $\mathcal{O}(N^2)$), and also would not work for variable-length sequences (i.e., arbitrary values of $N$). Instead, we define block-diagonal matrices $\boldsymbol{B}_l$, where each block is a repeated $H \times H$ lower-triangular matrix with strictly positive diagonal entries. Computing $\boldsymbol{B}_l^{-1}$ takes $\mathcal{O}(H^2)$, and multiplication by $\boldsymbol{B}_l$ or $\boldsymbol{B}_l^{-1}$ can be done in $\mathcal{O}(NH)$ in parallel. We stack $L$ such matrices $\boldsymbol{B}_l$ and define the triangular map $\boldsymbol{F} = \boldsymbol{C} \circ \boldsymbol{\Phi}_2 \circ \boldsymbol{B}_L \circ \cdots \circ \boldsymbol{B}_1 \circ \boldsymbol{\Phi}_1 \circ \boldsymbol{D} \circ \boldsymbol{\Lambda}$. The blocks in every other layer are shifted by an offset $H/2$ to let the model capture long-range dependencies. Note that now we use two element-wise learnable splines $\boldsymbol{\Phi}_1$ and $\boldsymbol{\Phi}_2$ before and after the block-diagonal layers. Figure 3 visualizes the overall sequence of maps and the Jacobians of each transformation. We name the temporal point process densities defined by the triangular map $\boldsymbol{F}$ as TriTPP.

Both the forward map $\boldsymbol{F}$ and its inverse $\boldsymbol{F}^{-1}$ can be evaluated in parallel in linear time, making TriTPP efficient for density computation and sampling. Our insight that TPP densities can be represented by increasing triangular maps was crucial for arriving at this result. Alternative representations of TriTPP, e.g., in terms of the compensator $\Lambda^*$ or the conditional intensity $\lambda^*$, are cumbersome and do not emphasize the parallelism of the model. TriTPP and our parametrizations of IPP, RP, MRP can be efficiently implemented on GPU to handle batches of variable-length sequences (Appendix C).

## 4 Differentiable sampling-based losses for temporal point processes

Fast parallel sampling allows us to efficiently answer prediction queries such as "How many events are expected to happen in the next hour given the history?". More importantly, it enables us to efficiently train TPP models using objective functions of the form $\mathbb{E}_p[g(\boldsymbol{t})]$. This includes using $p(\boldsymbol{t})$ to specify the policy in reinforcement learning [17], to impute missing data during training [11] or to define an approximate posterior in variational inference (Section 4.2). In all but trivial cases the expression $\mathbb{E}_p[g(\boldsymbol{t})]$ has no closed form, so we need to estimate its gradients w.r.t. the parameters of $p(\boldsymbol{t})$ using Monte Carlo (MC). Recall that we can sample from $p(\boldsymbol{t})$ by applying the map $\boldsymbol{F}^{-1}$ to $\boldsymbol{z}$ drawn from an HPP with unit rate. This enables the so-called reparametrization trick [18]. Unfortunately, this is not enough. Sampling-based losses for TPPs are in general not differentiable. This is a property of the loss functions that is independent of the parametrization of $p(\boldsymbol{t})$ or the sampling method. In the following, we provide a simple example and a solution to this problem.

### 4.1 Entropy maximization

Consider the problem of maximizing the entropy of a TPP. An entropy penalty can be used as a regularizer during density estimation [19] or as a part of the ELBO in variational inference. Let $p_\lambda(\boldsymbol{t})$ be a homogeneous Poisson process on $[0, T]$ with rate $\lambda > 0$. It is known that the entropy is maximized when $\lambda = 1$ [20], but for sake of example assume that we want to learn $\lambda$ that maximizes the entropy $-\mathbb{E}_p[\log p_\lambda(\boldsymbol{t})]$ with gradient ascent. We sample from $p_\lambda(\boldsymbol{t})$ by drawing a sequence $\boldsymbol{z} = (z_1, z_2, ...)$ from a HPP with unit rate and applying the inverse map $\boldsymbol{t} = \boldsymbol{F}_\lambda^{-1}(\boldsymbol{z}) = \frac{1}{\lambda} \boldsymbol{z}$

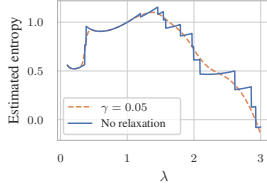

Figure 4: Monte Carlo estimate of the entropy.

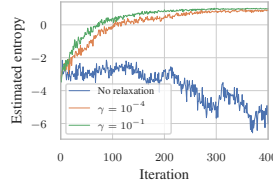

Figure 5: Maximizing the entropy with different values of $\gamma$.

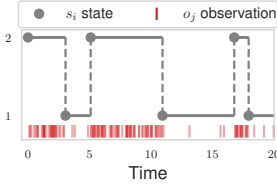

Figure 6: Markov modulated Poisson process with 2 states.

(Figure 2). We obtain an MC estimate of the entropy using a single such sample $\boldsymbol{t} = (t_1, t_2, ...)$ as

$$-\mathbb{E}_p[\log p_\lambda(\boldsymbol{t})] \approx \lambda T - \sum_{i=1}^{\infty} \mathbf{1}(t_i \leq T) \log \lambda = \lambda T - \sum_{i=1}^{\infty} \mathbf{1}\left(\frac{1}{\lambda} z_i \leq T\right) \log \lambda \qquad (3)$$

Here, the indicator function $\mathbf{1}(\cdot)$ discards all the events $t_i > T$. We can see that for any sample $\boldsymbol{z}$ the right-hand side of Equation 3 is not continuous w.r.t. $\lambda$ at points $\lambda = \frac{1}{T} z_i$. At such points, decreasing $\lambda$ by an infinitesimal amount will "push" the sample $t_i = \frac{1}{\lambda} z_i$ outside the $[0, T]$ interval, thus increasing $\log p_\lambda(\boldsymbol{t})$ by a constant $\log \lambda$. We plot the right-hand side of Equation 3 as a function of $\lambda$ in Figure 4, estimated with 5 MC samples. Clearly, such function cannot be optimized with gradient ascent. Increasing the number of MC samples almost surely adds more points of discontinuity and does not fix the problem. In general, non-differentiability arises when estimating expectations of a function $g(\boldsymbol{t})$ that depends on the events $t_i$ inside $[0, T]$. For any TPP density $p(\boldsymbol{t})$, the discontinuities occur at the parameter values that map the HPP realizations $z_i$ exactly to the interval boundary $T$.

**Relaxation.** We obtain a differentiable approximation to Equation 3 by relaxing the indicator functions as $\mathbf{1}(t_i \leq T) \approx \sigma_\gamma(T - t_i)$, where $\sigma_\gamma(x) = 1/(1 + \exp(-x/\gamma))$ is the sigmoid function with a temperature parameter $\gamma > 0$. Decreasing the temperature $\gamma$ makes the approximation more accurate, but complicates optimization, similarly to the Gumbel-softmax trick [21]. Figure 5 shows convergence plots for different values of $\gamma$. Our relaxation applies to MC estimation of any function $g(\boldsymbol{t})$ that can be expressed in terms of the indicator functions. This method also enables differentiable sampling with reparametrization from a Poisson distribution, which might be of independent interest.

## 4.2 Variational inference for Markov jump processes

Combining fast sampling (Section 3) with the differentiable relaxation opens new applications for TPPs. As an example, we design a variational inference scheme for Markov jump processes.

**Background.** A Markov jump process (MJP) $\{s(t)\}_{t \geq 0}$ is a piecewise-constant stochastic process on $\mathbb{R}_+$. At any time $t$, the process occupies a discrete state $s(t) \in \{1, ..., K\}$. The times when the state changes are called jumps. A trajectory of an MJP on an interval $[0, T]$ with $N$ jumps can be represented by a tuple $(\boldsymbol{t}, \boldsymbol{s})$ of jump times $\boldsymbol{t} = (t_1, ..., t_N)$ and the visited states $\boldsymbol{s} = (s_1, ..., s_{N+1})$. Note that $N$ may vary for different trajectories. The prior over the trajectories $p(\boldsymbol{t}, \boldsymbol{s}|\boldsymbol{\pi}, \boldsymbol{A})$ of an MJP is governed by an initial state distribution $\boldsymbol{\pi}$ and a $K \times K$ generator matrix $\boldsymbol{A}$ (see Appendix B.1).

MJPs are commonly used to model the unobserved (latent) state of a system. In a latent MJP, the state $s(t)$ influences the behavior of the system and indirectly manifests itself via some observations $\boldsymbol{o}$. For concreteness, we consider the Markov-modulated Poisson process (MMPP) [22]. In an MMPP, each of the $K$ states of the MJP has an associated observation intensity $\lambda_k$. An MMPP is an inhomogeneous Poisson process where the intensity depends on the current MJP state as $\lambda(t) = \lambda_{s(t)}$. For instance, a 2-state MMPP can model the behavior of a social network user, who switches between an "active" (posting a lot) and "inactive" (working or sleeping) states (Figure 6). Given the observations $\boldsymbol{o}$, we might be interested in inferring the trajectory $(\boldsymbol{t}, \boldsymbol{s})$, the model parameters $\boldsymbol{\theta} = \{\boldsymbol{\pi}, \boldsymbol{A}, \boldsymbol{\lambda}\}$, or both.

**Variational inference.** The posterior distribution $p(\boldsymbol{t}, \boldsymbol{s}|\boldsymbol{o}, \boldsymbol{\theta})$ of MMPP is intractable, so we approximate it with a variational distribution $q(\boldsymbol{t}, \boldsymbol{s}) = q(\boldsymbol{t})q(\boldsymbol{s}|\boldsymbol{t})$. Note that this is *not* a mean-field approximation used in other works [23]. We model the distribution over the jump times $q(\boldsymbol{t})$ with TriTPP (Section 3.3). We find the best approximate posterior by maximizing the ELBO [24]

$$\max_{q(\boldsymbol{t})} \max_{q(\boldsymbol{s}|\boldsymbol{t})} \mathbb{E}_{q(\boldsymbol{t})}\left[\mathbb{E}_{q(\boldsymbol{s}|\boldsymbol{t})}[\log p(\boldsymbol{o}|\boldsymbol{t}, \boldsymbol{s}, \boldsymbol{\theta}) + \log p(\boldsymbol{t}, \boldsymbol{s}|\boldsymbol{\theta}) - \log q(\boldsymbol{t}, \boldsymbol{s})]\right] \qquad (4)$$

Given jump times $\boldsymbol{t}$, the true posterior over the states $p(\boldsymbol{s}|\boldsymbol{t}, \boldsymbol{o}, \boldsymbol{\theta})$ is just the posterior of a discrete hidden Markov model (HMM). This means that we only need to model $q(\boldsymbol{t})$; the optimal $q^\star(\boldsymbol{s}|\boldsymbol{t})$, i.e.

$$q^\star(\boldsymbol{s}|\boldsymbol{t}) = \underset{q(\boldsymbol{s}|\boldsymbol{t})}{\arg\max}\, \mathbb{E}_{q(\boldsymbol{s}|\boldsymbol{t})}\left[\log p(\boldsymbol{o}|\boldsymbol{t}, \boldsymbol{s}, \boldsymbol{\theta}) + \log p(\boldsymbol{t}, \boldsymbol{s}|\boldsymbol{\theta}) - \log q(\boldsymbol{s}|\boldsymbol{t})\right] = p(\boldsymbol{s}|\boldsymbol{t}, \boldsymbol{o}, \boldsymbol{\theta}) \quad (5)$$

can be found by doing inference in an HMM — doable efficiently via the forward-backward algorithm [25]. The inner expectation w.r.t. $q(\boldsymbol{s}|\boldsymbol{t})$ in Equation 4 can be computed analytically. We approximate the expectation w.r.t. $q(\boldsymbol{t})$ with Monte Carlo. Since all terms of Equation 4 are not differentiable, we apply our relaxation from Section 4.1. We provide a full derivation of the ELBO and the implementation details in Appendix B.2.

The proposed framework is not limited to approximating the posterior over the trajectories. With small modifications (Appendix B.3), we can simultaneously learn the parameters $\boldsymbol{\theta}$, either obtaining a point estimate $\boldsymbol{\theta}^\star$ or a full approximate posterior $q(\boldsymbol{\theta})$. Our variational inference scheme can also be extended to other continuous-time discrete-state models, such as semi-Markov processes [26].

## 5 Related work

**Triangular maps** [2] can be seen as a generalization of autoregressive normalizing flows [27, 28, 15]. Existing normalizing flow models are either limited to fixed-dimensional data [29, 30] or are inherently sequential [31, 32]. Our model proposed in Section 3.3 can handle variable-length inputs, and allows for both $\boldsymbol{F}$ and $\boldsymbol{F}^{-1}$ to be evaluated efficiently in parallel.

**Sampling from TPPs.** Inverse method for sampling from inhomogeneous Poisson processes can be dated back to Çinlar [33]. However, traditional inversion methods for IPPs are different from our approach (Section 3). First, they are typically sequential. Second, existing methods either use extremely basic compensators $\Lambda(t)$, such as $\lambda t$ or $e^{\alpha t}$, or require numerical inversion [34]. As an alternative to inversion, thinning approaches [35] became the dominant paradigm for generating IPPs, and TPPs in general. Still, sampling via thinning has a number of disadvantages. Thinning requires a piecewise-constant upper bound on $\lambda(t)$, which might not always be easy to find. If the bound is not tight, a large fraction of samples will be rejected. Moreover, thinning is not differentiable, doesn't permit reparametrization, and is hard to express in terms of parallel operations on tensors [36]. Our inversion-based sampling addresses all the above limitations. It's also possible to generate an IPP by first drawing $N \sim \text{Poisson}(\Lambda(T))$ and then sampling $N$ points $t_i$ i.i.d. from a density $p(t) = \lambda(t)/\Lambda(T)$ [37]. Unlike inversion, this method is only applicable to Poisson processes. Also, the operation of sampling $N$ is not differentiable, which limits the utility of this approach.

**Inhomogeneous Poisson processes** are commonly defined by specifying the intensity function $\lambda(t)$ via a latent Gaussian process [38]. Such models are flexible, but highly intractable. It's possible to devise approximations by, e.g., bounding the intensity function [39, 40]. Our spline parametrization of IPP compares favorably to the above models: it is also highly flexible, has a tractable likelihood and places no restrictions on the intensity. Importantly, it is much easier to implement and train. If uncertainty is of interest, we can perform approximate Bayesian inference on the spline coefficients [24]. Recently, Morgan et al. [41] used splines to model the intensity function of IPPs. Since $\Lambda^{-1}$ cannot be computed analytically for their model, sampling via thinning is the only available option.

**Modulated renewal processes** have been known for a long time [13, 42], but haven't become as popular as IPPs among practitioners. This is not surprising, since inference and sampling in MRPs are even more challenging than in Cox processes [43, 44]. Our proposed parametrization addresses the shortcomings of existing approaches and makes MRPs straightforward to apply in practice.

**Neural TPPs.** Du et al. [1] proposed a TPP model based on a recurrent neural network. Follow-up works improved the flexibility of RNN-based TPPs by e.g. changing the RNN architecture [45], using more expressive conditional hazard functions [10, 46] or modeling the inter-event time distribution with normalizing flows [11]. All the above models are inherently sequential and therefore inefficient for sampling (Section 6.1). Recently, Turkmen et al. [36] proposed to speed up RNN-based *marked* TPPs by discretizing the interval $[0, T]$ into a regular grid. Samples within each grid cell can be produced in parallel for each mark, but the cells themselves still must be processed sequentially.

**Latent space models.** TPPs governed by latent Markov dynamics have intractable likelihoods that require approximations [47, 48]. For MJPs, the state-of-the-art approach is the Gibbs sampler by Rao & Teh [49]. It allows to exactly sample from the posterior $p(\boldsymbol{t}, \boldsymbol{s}|\boldsymbol{o}, \boldsymbol{\theta})$, but is known to converge

slowly if the parameters $\boldsymbol{\theta}$ are to be learned as well [50]. Existing variational inference approaches for MJPs can only learn a fixed time discretization [23] or estimate the marginal statistics of the posterior [51, 52]. In contrast, our method (Section 4.2) produces a full distribution over the jump times.

## 6 Experiments

### 6.1 Scalability

**Setup.** The key feature of TriTPP is its ability to compute likelihood and generate samples in parallel, which is impossible for RNN-based models. We quantify this difference by measuring the runtime of the two models. We implemented TriTPP and RNN models in PyTorch [53]. The architecture of the RNN model is nearly identical to the ones used in [1, 10, 11], except that the cumulative conditional hazard function is parametrized with a spline [3] to enable closed-form sampling. Appendix E contains the details for this and other experiments. We measure the runtime of (a) computing the log-likelihood (and backpropagate the gradients) for a batch of 100 sequences of varying lengths and (b) sample sequences of the same sizes. We used a machine with an Intel Xeon E5-2630 v4 @ 2.20 GHz CPU, 256GB RAM and an Nvidia GTX1080Ti GPU. The results are averaged over 100 runs.

**Results.** Figure 7 shows the runtimes for varying sequence lengths. Training is rather fast for both models, on average taking 1-10ms per iteration. RNN is slightly faster for short sequences, but is outperformed by TriTPP on sequences with more than 400 events. Note that during training we used a highly optimized RNN implementation based on custom CUDA kernels (since all the event times $t_i$ are already known). In contrast, TriTPP is implemented using generic PyTorch operations. When it comes to sampling, we notice a massive gap in performance between TriTPP and the RNN model. This happens because RNN-based TPPs are defined autoregressively and can only produce samples $t_i$ one by one: to obtain $p(t_i|t_1, ..., t_{i-1})$ we must know all the past events. Recently proposed transformer TPPs [54, 55] are defined in a similar autoregressive way, so they are likely to be as slow for sampling as RNNs. TriTPP generates all the events in a sequence in parallel, which makes it more than 100 times faster than the recurrent model for longer sequences.

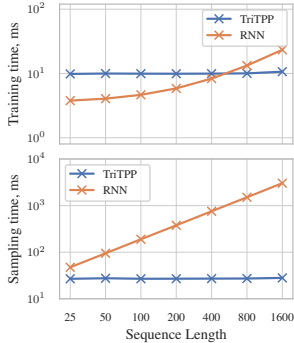

Figure 7: Scalability analysis. Standard devs. are below 1ms.

### 6.2 Density estimation

**Setup.** A fast TPP model is of little use if it cannot accurately learn the data distribution. The main goal of this experiment is to establish whether TriTPP can match the flexibility of RNN-based TPPs. As baselines, we use the IPP, RP and MRP models from Section 3.2 and Hawkes process [56].

**Datasets.** We use 6 synthetic datasets from Omi et al. [10]: Hawkes1&2 [7], self-correcting (SC) [16], inhomogeneous Poisson (IPP), renewal (RP) and modulated renewal (MRP) processes. Note that the data generators for IPP, RP and MRP by Omi et al. are *not* parametrized using splines, so these datasets are not guaranteed to be fitted perfectly by our models. We also consider 7 real-world datasets: PUBG (online gaming), Reddit-Comments, Reddit-Submissions (online discussions), Taxi (customer pickups), Twitter (tweets) and Yelp1&2 (check-in times). See Appendix D for more details.

**Metrics.** The standard metric for comparing generative models, including TPPs, is negative log-likelihood (NLL) on a hold-out set [36, 10, 11]. We partitioned the sequences in each dataset into train/validation/test sequences (60%/20%/20%). We trained the models by minimizing the NLL of the train set using Adam [57]. We tuned the following hyperparameters: $L_2$ regularization $\{0, 10^{-5}, 10^{-4}, 10^{-3}\}$, number of spline knots $\{10, 20, 50\}$, learning rate $\{10^{-3}, 10^{-2}\}$, hidden size $\{32, 64\}$ for RNN, number of blocks $\{2, 4\}$ and block size $\{8, 16\}$ for TriTPP. We used the validaiton set for hyperparameter tuning, early stopping and model development. We computed the results for the test set only once before including them in the paper. All results are averaged over 5 runs.

While NLL is a popular metric, it has known failure modes [58]. For this reason, we additionally computed maximum mean discrepancy (MMD) [59] between the test sets and the samples drawn from each model after training. To measure similarity between two realizations $\boldsymbol{t}$ and $\boldsymbol{t}'$, we use a Gaussian kernel $k(\boldsymbol{t}, \boldsymbol{t}') = \exp(-d(\boldsymbol{t}, \boldsymbol{t}')/2\sigma^2)$, where $d(\boldsymbol{t}, \boldsymbol{t}')$ is the "counting measure" distance from [60,

Table 1: Average test set NLL on synthetic and real-world datasets (lower is better). Best NLL in **bold**, second best underlined. Results with standard deviations can be found in Appendix F.1.

| | Hawkes1 | Hawkes2 | SC | IPP | MRP | RP | PUBG | Reddit-C | Reddit-S | Taxi | Twitter | Yelp1 | Yelp2 |
|---|---|---|---|---|---|---|---|---|---|---|---|---|---|
| IPP | 1.06 | 1.03 | 1.00 | **0.71** | 0.70 | 0.89 | -0.06 | -1.59 | -4.08 | **-0.68** | 1.60 | 0.62 | -0.05 |
| RP | 0.65 | 0.08 | 0.94 | 0.85 | 0.68 | **0.24** | 0.12 | -2.08 | -4.00 | -0.58 | 1.20 | 0.67 | -0.02 |
| MRP | 0.65 | 0.07 | 0.93 | 0.71 | 0.36 | 0.25 | -0.83 | -2.13 | -4.38 | **-0.68** | 1.23 | 0.61 | **-0.10** |
| Hawkes | **0.51** | 0.06 | 1.00 | 0.86 | 0.98 | 0.39 | 0.11 | **-2.40** | -4.19 | -0.64 | **1.04** | 0.69 | 0.01 |
| RNN | 0.52 | **-0.03** | **0.79** | 0.73 | 0.37 | **0.24** | -1.96 | **-2.40** | **-4.89** | -0.66 | 1.08 | 0.67 | -0.08 |
| TriTPP | 0.56 | 0.00 | 0.83 | **0.71** | 0.35 | **0.24** | **-2.41** | -2.36 | -4.49 | -0.67 | 1.06 | 0.64 | -0.09 |

Table 2: MMD between the hold-out test set and the generated samples (lower is better).

| | Hawkes1 | Hawkes2 | SC | IPP | MRP | RP | PUBG | Reddit-C | Reddit-S | Taxi | Twitter | Yelp1 | Yelp2 |
|---|---|---|---|---|---|---|---|---|---|---|---|---|---|
| IPP | 0.08 | 0.09 | 0.58 | **0.02** | 0.15 | 0.07 | **0.01** | 0.10 | 0.21 | 0.10 | 0.16 | 0.15 | 0.16 |
| RP | 0.06 | 0.06 | 1.13 | 0.34 | 1.24 | **0.01** | 0.46 | 0.07 | 0.18 | 0.57 | 0.14 | 0.16 | 0.23 |
| MRP | 0.05 | 0.06 | 0.50 | **0.02** | 0.11 | 0.02 | 0.12 | 0.09 | 0.20 | 0.09 | 0.13 | 0.13 | 0.16 |
| Hawkes | 0.02 | 0.04 | 0.58 | 0.36 | 0.65 | 0.05 | 0.16 | **0.04** | 0.35 | 0.20 | 0.20 | 0.20 | 0.32 |
| RNN | **0.01** | **0.02** | **0.19** | 0.09 | 0.17 | **0.01** | 0.23 | **0.04** | **0.09** | 0.13 | **0.08** | 0.19 | 0.18 |
| TriTPP | 0.03 | 0.03 | 0.23 | **0.02** | **0.08** | **0.01** | 0.16 | 0.07 | 0.16 | **0.08** | **0.08** | 0.12 | **0.14** |

Equation 3]. For completeness, we provide the definitions in Appendix E.2. MMD quantifies the dissimilarity between the true data distribution $p^\star(\boldsymbol{t})$ and the learned density $p(\boldsymbol{t})$ — lower is better.

**Results.** Table 1 shows the test set NLLs for all models and datasets. We can see that the RNN model achieves excellent scores and outperforms the simpler baselines, which is consistent with earlier findings [1]. TriTPP is the only method that is competitive with the RNN — our method is within 0.05 nats of the best score on 11 out of 13 datasets. TriTPP consistently beats MRP, RP and IPP, which confirms that learnable block-diagonal transformations improve the flexibility of the model. The gap get larger on the datasets such as Hawkes, SC, PUBG and Twitter, where the inability of MRP to learn self-exciting and self-correcting behavior is especially detrimental. While Hawkes process is able to achieve good scores on datasets with "bursty" event occurrences (Reddit, Twitter), it is unable to adequately model other types of behavior (SC, MRP, PUBG).

Table 2 reports the MMD scores. The results are consistent with the previous experiment: models with lower NLL typically obtain lower MMD. One exception is the Hawkes process that achieves low NLL but high MMD on Taxi and Twitter. TriTPP again consistently demonstrates excellent performance. Note that MMD was computed using the test sequences that were unseen during training. This means that TriTPP models the data distribution better than other methods, and does not just simply overfit the training set. In Appendix F.1, we provide additional experiments for quantifying the quality of the distributions learned by different models. Overall, we conclude that TriTPP is flexible and able to model complex densities, in addition to being significantly more efficient than RNN-based TPPs.

### 6.3 Variational inference

**Setup.** We apply our variational inference method (Section 4.2) for learning the posterior distribution over the latent trajectories of an MMPP. We simulate an MMPP with $K = 3$ latent states. As a baseline, we use the state-of-the-art MCMC sampler by Rao & Teh [49].

**Results.** Figure 8 shows the true latent MJP trajectory, as well as the marginal posterior probabilities learned by our method and the MCMC sampler of Rao & Teh. We can see that TriTPP accurately recovers the true posterior distribution over the trajectories. The two components that enable our new variational inference approach are our efficient parallel sampling algorithm for TriTPP (Section 3) and the differential relaxation (Section 4). Appendix F.2 contains an additional experiment on real-world data, where we both learn the parameters $\boldsymbol{\theta}$ and infer the posterior over the trajectories.

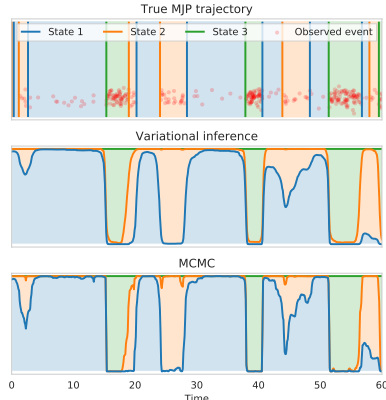

Figure 8: Posterior distributions over the latent trajectory of an MMPP learned using our VI approach & MCMC.

# 7 Future work & conclusions

**Future work & limitations.** We parametrized the nonlinear transformations of our TPP models with splines. Making a spline more flexible requires increasing the number of knots, which increases the number of parameters and might lead to overfitting. New deep *analytically invertible* functions will improve both our models, as well as normalizing flows in general. Currently, TriTPP is not applicable to marked TPPs [5]. Extending our model to this setting is an important task for future work.

**Conclusions.** We have shown that TPP densities can be represented with increasing triangular maps. By directly parametrizing the respective transformations, we are able to construct TPP models, for which both density evaluation and sampling can be done efficiently in parallel. Using the above framework, we defined TriTPP— a new class of flexible probability distributions over variable-length sequences. In addition to being highly efficient thanks to its parallelism, TriTPP shows excellent performance on density estimation, as shown by our experiments. High flexibility and efficiency of TriTPP allow it to be used as a plug-and-play component of other machine learning models.

## Broader impact

Existing works have applied TPPs and MJPs for analyzing electronic health records [61, 62], detecting anomalies in network traffic [63, 64] and modeling user behavior on online platforms [65, 66]. Thanks to fast sampling, our model can be used for solving new prediction tasks on such data, and the overall improved scalability allows practitioners to work with larger datasets. We do not find any of the above use cases ethically questionable, though, general precautions must be implemented when handling sensitive personal data. Since our model exploits fast parallel computations, has fewer parameters and converges in fewer iterations, it is likely to be more energy-efficient compared to RNN-based TPPs. However, we haven't performed experiments analyzing this specific aspect of our model.

## Acknowledgments

This research was supported by the German Federal Ministry of Education and Research (BMBF), grant no. 01IS18036B, the Software Campus Project Deep-RENT and by the BMW AG. The authors of this work take full responsibilities for its content.

## Footnotes

[1]For convenience, we provide a list of abbreviations and notation used in the paper in Appendix A.

[2]Note that some other works instead define $\boldsymbol{F}$ as the map that pushes the density $\tilde{p}(\boldsymbol{z})$ into $p(\boldsymbol{x})$.

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
