[Supplementary Material]

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

[3]`https://kaggle.com/skihikingkevin/pubg-match-deaths`

[4]`https://pushshift.io/`

[5]`https://www.kaggle.com/c/nyc-taxi-trip-duration/data`

[6]`https://twitter.com`

[7]`https://www.yelp.com/dataset/challenge`

[8]`https://www.kaggle.com/shawon10/web-log-dataset`

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

# A    Abbreviations and notation

Abbreviations:

- RNN — recurrent neural network
- TPP — temporal point process
- HPP — homogeneous Poisson process (compensator is $\Lambda^*(t) = \lambda t$ for some $\lambda > 0$)
- IPP — inhomogeneous Poisson process
- RP — renewal process
- MRP — modulated renewal process
- MJP – Markov jump process
- MMPP — Markov modulated Poisson process
- MC — Monte Carlo
- VI — variational inference

Table 3: Notation used throughout the paper.

| Notation | Description |
|---|---|
| $\boldsymbol{t} = (t_1, ..., t_N)$ | Variable-length realization of a TPP. |
| $p(\boldsymbol{t})$ | Density of a point process, also called likelihood (Equation 1). |
| $\lambda^*(t) = \lambda(t\|t_1, ..., t_{i-1})$ | Conditional intensity at time $t$, where $t_{i-1}$ is the last event before $t$. |
| $\Lambda^*(t) = \Lambda(t\|t_1, ..., t_{i-1})$ $= \int_0^T \lambda^*(u)du$ | Cumulative conditional intensity at time $t$, also known as the compensator. |
| $\Lambda(t)$ | (Unconditional) cumulative intensity of a Poisson process. |
| $\Phi(\tau)$ | Cumulative hazard function of a renewal process. |
| $\boldsymbol{C}$ | The $N \times N$ cumulative sum matrix, $C_{ij} = \begin{cases} 1 & \text{if } i \leq j, \\ 0 & \text{else.} \end{cases}$ |
| $\boldsymbol{D} \equiv \boldsymbol{C}^{-1}$ | The $N \times N$ difference matrix, $D_{ij} = \begin{cases} 1 & \text{if } i = j, \\ -1 & \text{if } i = j + 1, \\ 0 & \text{else.} \end{cases}$ |
| $\boldsymbol{F} = (f_1, ..., f_N)$ | Increasing lower-triangular map that converts a realization $\boldsymbol{t}$ of an arbitrary TPP with compensator $\Lambda^*$ into a sample $\boldsymbol{z}$ from an HPP with unit rate. |
| $f_i(t_1, ..., t_i) = \Lambda(t_i\|t_1, ..., t_{i-1})$ | Component function of $\boldsymbol{F}$. |
| $\mathbf{1}(x)$ | Indicator function, $\mathbf{1}(x) = \begin{cases} 1 & \text{if } x \text{ is True,} \\ 0 & \text{else.} \end{cases}$ |
| $\gamma$ | Temperature parameter for the diff. relaxation (Section 4.1). |

# B  Variational inference for Markov jump processes

## B.1  Generative model for MJP and MMPP

**Markov jump process.** We represent the trajectory of an MJP as a tuple $(\boldsymbol{t}, \boldsymbol{s})$, where $\boldsymbol{t} = (t_1, ..., t_N)$ are the (strictly increasing) jump times and $\boldsymbol{s} = (s_1, ..., s_{N+1})$ is the sequence of visited states. For convenience, we additionally set $t_0 = 0$ and $t_{N+1} = T$.

The distribution over the trajectories $(\boldsymbol{t}, \boldsymbol{s})$ is defined by a $K \times K$ generator matrix $\boldsymbol{A}$ and an initial state distribution $\boldsymbol{\pi}$. Each entry $A_{kl} \in \mathbb{R}_+$ denotes the rate of transition from state $k$ to state $l$ of the the MJP. Note that we use the formulation that permits self-jumps [67] (i.e., it may happen that $s_i = s_{i+1}$). We can denote the total transition rate of state $s_i$ as $A_{s_i} = \sum_{k=1}^{K} A_{s_i k}$. We can simulate an MJP trajectory using the following procedure

$$
\begin{aligned}
s_1 &\sim \text{Categorical}(\boldsymbol{\pi}) \\
t_i - t_{i-1} &=: \tau_i \sim \text{Exponential}\,(A_{s_i}) \\
s_{i+1} &\sim \text{Categorical}\,(\boldsymbol{A}_{s_i:}/A_{s_i})
\end{aligned}
\tag{6}
$$

Here, $\boldsymbol{A}_{s_i:}/A_{s_i}$ is the $s_i$'th row of $\boldsymbol{A}$ that is normalized to sum up to 1.

The likelihood of a trajectory $(\boldsymbol{t}, \boldsymbol{s})$ for an MJP with parameters $(\boldsymbol{\pi}, \boldsymbol{A})$ can be computed as

$$
p(\boldsymbol{t}, \boldsymbol{s}|\boldsymbol{\pi}, \boldsymbol{A}) = \pi_{s_1} \left( \prod_{i=1}^{N} A_{s_{i-1}s_i} \right) \exp \left( -\sum_{i=1}^{N+1} (t_i - t_{i-1}) \sum_{l=1}^{K} A_{s_i l} \right)
$$

We reformulate this expression using indicators $\mathbf{1}(\cdot)$, which will make the ELBO computation easier

$$
= \left( \prod_{k=1}^{K} \pi_k^{\mathbf{1}(s_1=k)} \right) \left( \prod_{i=1}^{N} \prod_{k=1}^{K} \prod_{l=1}^{K} A_{kl}^{\mathbf{1}(s_i=k, s_{i+1}=l)} \right) \exp \left( -\sum_{i=1}^{N+1} (t_i - t_{i-1}) \sum_{k=1}^{K} \mathbf{1}(s_i = k) \left( \sum_{l=1}^{K} A_{kl} \right) \right)
$$

By applying the logarithm to the above equation, we obtain

$$
\begin{aligned}
\log p(\boldsymbol{t}, \boldsymbol{s}|\boldsymbol{\pi}, \boldsymbol{A}) = &\left( \sum_{k=1}^{K} \mathbf{1}(s_1 = k) \log \pi_k \right) + \left( \sum_{i=1}^{N} \sum_{k=1}^{K} \sum_{l=1}^{K} \mathbf{1}(s_i = k, s_{i+1} = l) \log A_{kl} \right) \\
&- \left( \sum_{i=1}^{N+1} (t_i - t_{i-1}) \sum_{k=1}^{K} \mathbf{1}(s_i = k) \left( \sum_{l=1}^{K} A_{kl} \right) \right)
\end{aligned}
\tag{7}
$$

**Markov modulated Poisson process.** Distribution of the observations $\boldsymbol{o} = (o_1, ..., o_M)$ of an MMPP depends on the latent MJP trajectory $(\boldsymbol{t}, \boldsymbol{s})$ and the rates of each state $\boldsymbol{\lambda} \in \mathbb{R}_+^K$. The observations $\boldsymbol{o}$ are sampled from an inhomogeneous Poisson process with piecewise-constant intensity that depends on the current state: $\lambda(t) = \lambda_{s(t)}$.

Likelihood of the observations $\boldsymbol{o}$ given $(\boldsymbol{t}, \boldsymbol{s})$ and $\boldsymbol{\lambda}$ can be computed as

$$
p(\boldsymbol{o}|\boldsymbol{t}, \boldsymbol{s}, \boldsymbol{\lambda}) = \left( \prod_{i=1}^{N+1} \lambda_{s_i}^{M_{[t_{i-1}, t_i)}} \right) \exp \left( -\sum_{i=1}^{N+1} (t_i - t_{i-1}) \lambda_{s_i} \right)
$$

where $M_{[t_{i-1}, t_i)}$ is the number of events $o_j$ in the interval $[t_{i-1}, t_i)$. Again, using indicator functions, we rewrite it as

$$
= \left( \prod_{i=1}^{N+1} \prod_{j=1}^{M} \left( \prod_{k=1}^{K} \lambda_k^{\mathbf{1}(s_i=k)} \right)^{\mathbf{1}(o_j \in [t_{i-1}, t_i))} \right) \exp \left( -\sum_{i=1}^{N+1} (t_i - t_{i-1}) \sum_{k=1}^{K} \mathbf{1}(s_i = k)\lambda_k \right)
$$

By applying the logarithm, we obtain

$$
\begin{aligned}
\log p(\boldsymbol{o}|\boldsymbol{t}, \boldsymbol{s}, \boldsymbol{\lambda}) = &\left( \sum_{i=1}^{N+1} \sum_{j=1}^{M} \mathbf{1}(o_j \in [t_{i-1}, t_i)) \sum_{k=1}^{K} \mathbf{1}(s_i = k) \log \lambda_k \right) \\
&- \left( \sum_{i=1}^{N+1} (t_i - t_{i-1}) \sum_{k=1}^{K} \mathbf{1}(s_i = k)\lambda_k \right)
\end{aligned}
\tag{8}
$$

## B.2 Derivation of the ELBO

The true posterior of an MMPP $p(\boldsymbol{t}, \boldsymbol{s}|\boldsymbol{o}, \boldsymbol{\pi}, \boldsymbol{A}, \boldsymbol{\lambda}) \propto p(\boldsymbol{t}, \boldsymbol{s}, \boldsymbol{o}|\boldsymbol{\pi}, \boldsymbol{A}, \boldsymbol{\lambda}) = p(\boldsymbol{t}, \boldsymbol{s}|\boldsymbol{\pi}, \boldsymbol{A})p(\boldsymbol{o}|\boldsymbol{t}, \boldsymbol{s}, \boldsymbol{\lambda})$ is intractable. We approximate it with a variational distribution $q(\boldsymbol{t}, \boldsymbol{s})$ by maximizing the evidence lower bound (ELBO) [24]

$$\text{ELBO}(q, \boldsymbol{\theta}) = \mathbb{E}_{q(\boldsymbol{t}, \boldsymbol{s})}\left[\underbrace{\log p(\boldsymbol{t}, \boldsymbol{s}|\boldsymbol{\pi}, \boldsymbol{A})}_{\text{trajectory log-likelihood}} + \underbrace{\log p(\boldsymbol{o}|\boldsymbol{t}, \boldsymbol{s}, \boldsymbol{\lambda})}_{\text{observations log-likelihood}} \underbrace{- \log q(\boldsymbol{t}, \boldsymbol{s})}_{\text{entropy}}\right]$$

Recall that we model the approximate posterior as $q(\boldsymbol{t}, \boldsymbol{s}) = q(\boldsymbol{t})q(\boldsymbol{s}|\boldsymbol{t})$, where $q(\boldsymbol{t})$ is defined using TriTPP and $q(\boldsymbol{s}|\boldsymbol{t})$ is evaluated exactly for each Monte Carlo sample $\boldsymbol{t}$. We rewrite the ELBO as

$$\text{ELBO}(q, \boldsymbol{\theta}) = \mathbb{E}_{q(\boldsymbol{t})}\left[\mathbb{E}_{q(\boldsymbol{s}|\boldsymbol{t})}\left[\log p(\boldsymbol{t}, \boldsymbol{s}|\boldsymbol{\pi}, \boldsymbol{A}) + \log p(\boldsymbol{o}|\boldsymbol{t}, \boldsymbol{s}, \boldsymbol{\lambda}) - \log q(\boldsymbol{s}|\boldsymbol{t})\right] - \log q(\boldsymbol{t})\right]$$

We already derived the expressions for $\log p(\boldsymbol{t}, \boldsymbol{s}|\boldsymbol{\pi}, \boldsymbol{A})$ (Equation 7) and $\log p(\boldsymbol{o}|\boldsymbol{t}, \boldsymbol{s}, \boldsymbol{\lambda})$ (Equation 8). The expression for $\log q(\boldsymbol{s}|\boldsymbol{t})$ can be obtained similarly

$$\log q(\boldsymbol{s}|\boldsymbol{t}) = \sum_{i=1}^{N+1}\sum_{k=1}^{K} \mathbf{1}(s_i = k) \log q(s_i = k|\boldsymbol{t}) \tag{9}$$

Finally, to compute the log-density $\log q(\boldsymbol{t})$ of a single sample $\boldsymbol{t} = (t_1, ..., t_N)$ we use the procedure described in Appendix C. We denote $\boldsymbol{z} = (z_1, ..., z_N, z_{N+1}) = \boldsymbol{F}(t_1, ..., t_N, T)$, and $z_{-1}$ is the last entry of $\boldsymbol{z}$.

$$\log q(\boldsymbol{t}) = \sum_{i=1}^{N} \log \left|\frac{\partial z_i}{\partial t_i}\right| - z_{-1} \tag{10}$$

**ELBO (non-differentiable version).** Putting everything together, we get

$$\begin{aligned}
\text{ELBO}(q, \boldsymbol{\theta}) = \mathbb{E}_{q(\boldsymbol{t})}\Bigg[\mathbb{E}_{q(\boldsymbol{s}|\boldsymbol{t})}\Bigg[&\sum_{k=1}^{K}\mathbf{1}(s_1 = k)\log\pi_k \\
&- \sum_{i=1}^{N+1}(t_i - t_{i-1})\sum_{k=1}^{K}\mathbf{1}(s_i = k)\sum_{l=1}^{K}A_{kl} \\
&+ \sum_{i=1}^{N}\sum_{k=1}^{K}\sum_{l=1}^{K}\mathbf{1}(s_i = k, s_{i+1} = l)\log A_{kl} \\
&+ \sum_{i=1}^{N+1}\sum_{j=1}^{M}\mathbf{1}(o_j \in [t_i, t_{i+1}))\sum_{k=1}^{K}\mathbf{1}(s_i = k)\log\lambda_k \\
&- \sum_{i=1}^{N+1}(t_i - t_{i-1})\sum_{k=1}^{K}\mathbf{1}(s_i = k)\lambda_k \\
&- \sum_{i=1}^{N+1}\sum_{k=1}^{K}\mathbf{1}(s_i = k)\log q(s_i = k|\boldsymbol{t})\Bigg] \\
&- \sum_{i=1}^{N}\log\left|\frac{\partial z_i}{\partial t_i}\right| + z_{-1}\Bigg]
\end{aligned} \tag{11}$$

Note that $N$ is the length of the sample $\boldsymbol{t}$ generated from $q(\boldsymbol{t})$, so $N$ will take different values for different samples. We can evaluate the inner expectation w.r.t. $q(\boldsymbol{s}|\boldsymbol{t})$ by using the following fact

$$\mathbb{E}_{q(\boldsymbol{s}|\boldsymbol{t})}\left[\mathbf{1}(s_i = k)\right] = q(s_i = k|\boldsymbol{t}) \quad \mathbb{E}_{q(\boldsymbol{s}|\boldsymbol{t})}\left[\mathbf{1}(s_i = k, s_{i+1} = l)\right] = q(s_i = k, s_{i+1} = l|\boldsymbol{t}) \tag{12}$$

Recall that we set $q(\boldsymbol{s}|\boldsymbol{t})$ to the true posterior $p(\boldsymbol{s}|\boldsymbol{t}, \boldsymbol{o}, \boldsymbol{\theta})$ over states given the jumps (Equation 5). This allows us to exactly compute both the posterior marginals $q(s_i = k|\boldsymbol{t})$ and the posterior

Figure 9: Examples of values involved in the ELBO computation.

transition probabilities $q(s_i = k, s_{i+1} = l | \boldsymbol{t})$ using the forward-backward algorithm [25, Equations 7, 8]. Therefore, the inner expectation w.r.t. $q(\boldsymbol{s}|\boldsymbol{t})$ in Equation 11 can be computed analytically.

**ELBO (differentiable relaxation).** The ELBO, as defined above in Equation 11, is discontinuous w.r.t. the parameters of the density $q(\boldsymbol{t})$ for the reasons described in Section 4.1. The expression inside the expectation depends only on the events $t_i$ that happen before $T$. Infinitesimal change in the parameters of $q(\boldsymbol{t})$ may "push" the point $t_i$ outside $[0, T]$, thus changing the function value by a fixed amount and resulting in a discontinuity.

We fix this problem using the approach described in Appendix C and Section 4.1. We obtain an "extended" sample $\tilde{\boldsymbol{t}}$ by first simulating a sequence $\tilde{\boldsymbol{z}} = (\tilde{z}_1, ..., \tilde{z}_{N'})$ from a HPP with unit rate and computing $\tilde{\boldsymbol{t}} = \boldsymbol{F}^{-1}(\tilde{\boldsymbol{z}})$ (more on this in Appendix C). We get a "clipped" / "padded" sample $\boldsymbol{t} = (t_1, ..., t_{N'})$ as $t_i = \min\{\tilde{t}_i, T\}$ (Figure 9). Finally, we compute $\boldsymbol{z} = (z_1, ..., z_{N'}) = \boldsymbol{F}(\boldsymbol{t})$ (this is necessary for computing the correct cumulative intensity $\Lambda^*(T)$ after clipping). We can now express the ELBO in terms of the "extended" samples $\tilde{\boldsymbol{t}}$ and "clipped" samples $\boldsymbol{t}$:

$$
\begin{aligned}
\text{ELBO}(q, \boldsymbol{\theta}) = \mathbb{E}_{q(\boldsymbol{t})}\Bigg[ \mathbb{E}_{q(\boldsymbol{s}|\boldsymbol{t})} \Bigg[ &\sum_{k=1}^{K} \mathbf{1}(s_1 = k) \log \pi_k \\
&- \sum_{i=1}^{N'} (t_i - t_{i-1}) \sum_{k=1}^{K} \mathbf{1}(s_i = k) \sum_{l=1}^{K} A_{kl} \\
&+ \sum_{i=1}^{N'} \mathbf{1}(\tilde{t}_i < T) \sum_{k=1}^{K} \sum_{l=1}^{K} \mathbf{1}(s_i = k, s_{i+1} = l) \log A_{kl} \\
&+ \sum_{i=1}^{N'} \sum_{j=1}^{M} \mathbf{1}(o_j \in [t_i, t_{i+1})) \sum_{k=1}^{K} \mathbf{1}(s_i = k) \log \lambda_k \\
&- \sum_{i=1}^{N'} (t_i - t_{i-1}) \sum_{k=1}^{K} \mathbf{1}(s_i = k) \lambda_k \\
&- \sum_{i=1}^{N'} \mathbf{1}(\tilde{t}_{i-1} < T) \sum_{k=1}^{K} \mathbf{1}(s_i = k) \log q(s_i = k | \boldsymbol{t}) \Bigg] \\
&- \sum_{i=1}^{N'} \mathbf{1}(\tilde{t}_i < T) \log \left| \frac{\partial z_i}{\partial t_i} \right| + z_{-1} \Bigg]
\end{aligned}
\tag{13}
$$

Changes from Equation 11 are highlighted in red. Even though the formula looks different, the result of evaluating Equation 13 will be *exactly* the same as for Equation 11. By using different notation we only made the process of "discarding" the events $t_i > T$ explicit. The new formulation allows us to obtain a differentiable relaxation. For this, we replace the indicator functions $\mathbf{1}(t_i < T)$ with

sigmoids $\sigma_\gamma(T - t_i)$. The indicator function $\mathbf{1}(o_j \in [t_i, t_{i+1}))$ can also be relaxed as

$$
\begin{aligned}
\mathbf{1}(o_j \in [t_i, t_{i+1})) &= \mathbf{1}(t_{i+1} > o_j) - \mathbf{1}(t_i \geq o_j) \\
&\approx \sigma_\gamma(t_{i+1} - o_j) - \sigma_\gamma(t_i - o_j)
\end{aligned}
\tag{14}
$$

By combining all these facts, we obtain a differentiable relaxation of the ELBO. Our method leads to an efficient implementation that uses batches of samples. We sample a batch of jump times $\{\mathbf{t}^{(1)}, \mathbf{t}^{(2)}, ...\}$ from $q(\mathbf{t})$, evaluate the posterior $q(\mathbf{s}|\mathbf{t})$ using with forward-backward for all of them in parallel, and evaluate the relaxed ELBO (Equation 13).

### B.3 Parameter estimation

In Section 4.2, we perform approximate posterior inference over the trajectories $(\mathbf{t}, \mathbf{s})$ by maximizing the ELBO w.r.t. $q(\mathbf{t}, \mathbf{s})$

$$
\max_{q(\mathbf{t}, \mathbf{s})} \mathbb{E}_q[\log p(\mathbf{t}, \mathbf{s}|\boldsymbol{\theta}) + \log p(\mathbf{o}|\mathbf{t}, \mathbf{s}, \boldsymbol{\theta}) - \log q(\mathbf{t}, \mathbf{s})]
\tag{15}
$$

Since $\mathrm{ELBO}(q, \boldsymbol{\theta})$ provides a lower bound on the marginal log-likelihood $\log p(\mathbf{o}|\boldsymbol{\theta})$, we can also simultaneously learn the model parameters $\boldsymbol{\theta} = \{\boldsymbol{\pi}, \mathbf{A}, \boldsymbol{\lambda}\}$ by solving the following optimization problem (subject to appropriate constraints on $\boldsymbol{\theta}$)

$$
\max_{\boldsymbol{\theta}} \max_{q(\mathbf{t}, \mathbf{s})} \mathbb{E}_q[\log p(\mathbf{t}, \mathbf{s}|\boldsymbol{\theta}) + \log p(\mathbf{o}|\mathbf{t}, \mathbf{s}, \boldsymbol{\theta}) - \log q(\mathbf{t}, \mathbf{s})]
\tag{16}
$$

Finally, we can perform fully Bayesian treatment and approximate the posterior distribution over the parameters as well as the trajectories. For this, we can place a prior $p(\boldsymbol{\theta})$ and approximate $p(\boldsymbol{\theta}, \mathbf{t}, \mathbf{s}|\mathbf{x})$ with $q(\boldsymbol{\theta}, \mathbf{t}, \mathbf{s}) = q(\boldsymbol{\theta})q(\mathbf{t})q(\mathbf{s}|\mathbf{t}, \boldsymbol{\theta})$. This corresponds to the following optimization problem

$$
\max_{q(\boldsymbol{\theta}, \mathbf{t}, \mathbf{s})} \mathbb{E}_q[\log p(\mathbf{t}, \mathbf{s}|\boldsymbol{\theta}) + \log p(\mathbf{o}|\mathbf{t}, \mathbf{s}, \boldsymbol{\theta}) - \log q(\mathbf{t}, \mathbf{s})] - \mathbb{KL}(q(\boldsymbol{\theta})\|p(\boldsymbol{\theta}))
\tag{17}
$$

where $\mathbb{KL}$ denotes KL-divergence. By applying our relaxation from Section 4, it's possible to solve all of the above optimization problems (Equations 15, 16, 17) using gradient ascent.

## C Implementation details

### C.1 Batch processing

By representing TPP densities with transformations, we can implement both (log-)density evaluation and sampling efficiently and in parallel. Our implementation enables parallelism not only for the events $t_i$ of a single sequence, but also for entire batches consisting of multiple sequences $\mathbf{t}$ of different length.

First, consider a single sequence $\mathbf{t} = (1, 2.5, 4)$ with $N = 3$ events, sampled from a TPP on the interval $[0, 5]$. We pad this sequence with $T = 5$, and additionally introduce a mask $\mathbf{m}$ that tells us which entries of the padded vector $\mathbf{t}$ correspond to actual events (i.e., not padding)

$$
\mathbf{t} = \begin{bmatrix} 1 & 2.5 & 4 & 5 \end{bmatrix} \qquad \mathbf{m} = \begin{bmatrix} 1 & 1 & 1 & 0 \end{bmatrix}
$$

We implement the transformation $\mathbf{F}$ (corresponding to the TPP density $p(\mathbf{t})$) similarly to normalizing flow frameworks like `torch.distributions` [53]. We define a method `forward` that computes $\mathbf{z}$, the result of the transformation, and $\mathbf{j}$, logarithm of the diagonal entries of the Jacobian $J_{\mathbf{F}}(\mathbf{t})$:

$$
\mathbf{z} = \mathbf{F}(\mathbf{t}) = \begin{bmatrix} z_1 & z_2 & z_3 & z_4 \end{bmatrix} \qquad \mathbf{j} = \begin{bmatrix} \log\left|\frac{\partial z_1}{\partial t_1}\right| & \log\left|\frac{\partial z_2}{\partial t_2}\right| & \log\left|\frac{\partial z_3}{\partial t_3}\right| & \log\left|\frac{\partial z_4}{\partial t_4}\right| \end{bmatrix}
$$

From the definition of $\mathbf{F}$ (Table 3), we can see that the last entry of $\mathbf{z}$ (that we denote as $z_{-1}$) corresponds to $\Lambda^*(T)$. Also, each entry $j_i$ of $\mathbf{j}$ corresponds to $\log\left|\frac{\partial \Lambda^*(t_i)}{\partial t_i}\right|$. Therefore, we can compute the log-density $\log p(\mathbf{t})$ as

$$
\log p(\mathbf{t}) = \mathtt{sum}(\mathbf{m} \odot \mathbf{j}) - z_{-1} = \sum_{i=1}^{N'} m_i \log\left|\frac{\partial z_i}{\partial t_i}\right| - z_{-1} = \sum_{i=1}^{N} \log\left|\frac{\partial \Lambda^*(t_i)}{\partial t_i}\right| - \Lambda^*(T)
\tag{18}
$$

where $N'$ denotes the length *with* the padding. We can verify that this is equal to the logarithm of the TPP density in Equation 1. Note that if we use a longer padding, such as

$$\boldsymbol{t} = \begin{bmatrix} 1 & 2.5 & 4 & 5 & 5 & 5 & 5 \end{bmatrix} \qquad \boldsymbol{m} = \begin{bmatrix} 1 & 1 & 1 & 0 & 0 & 0 & 0 \end{bmatrix}$$

then Equation 18 will still correctly compute the log-likelihood for the sequence. This observation allows to process multiple sequences $\{\boldsymbol{t}^{(1)}, \boldsymbol{t}^{(2)}, ...\}$ in a single batch. We simply pad all the sequences with $T$ up to the length of the longest sequence, stack them into a matrix of shape [`batch_size`, `max_seq_len`] and process all of them in parallel.

As described in Section 3.3, we actually define $\boldsymbol{F}$ by stacking multiple transformations. We sequentially call the `forward` method for each transformation in the chain to obtain the final $\boldsymbol{z}$, and sum up the log-diagonals of the Jacobians $\boldsymbol{j}$ along the way. Each transformation and its Jacobian can be evaluated in parallel in linear time, making the whole operation efficient.

## C.2   Sampling

Sampling is implemented similarly. We start by simulating a vector $\tilde{\boldsymbol{z}}$ from a homogeneous Poisson process with unit rate. The length of $\tilde{\boldsymbol{z}}$ must be "long enough" (more on this later). We define the method `inverse` that computes $\tilde{\boldsymbol{t}} = \boldsymbol{F}^{-1}(\tilde{\boldsymbol{z}})$. We obtain a final sample $\boldsymbol{t}$ by clipping the entries of $\tilde{\boldsymbol{t}}$ as $t_i = \min\{\tilde{t}_i, T\}$. If we would like to compute the density of the generated sample $\boldsymbol{t}$, we will also need the mask $\boldsymbol{m}$ that can be obtained as $m_i = \mathbf{1}(\tilde{t}_i < T)$. In some use cases, such as entropy maximization (Section 4.1) or variational inference (Appendix B), we need to use a differentiable approximation to the mask $m_i = \sigma_\gamma(T - \tilde{t}_i)$. This recovers our relaxation from Section 4.1.

By slightly abusing the notation, we use $N'$ to denote the number of events in our initial HPP sample $\tilde{\boldsymbol{z}} = (\tilde{z}_1, ..., \tilde{z}_{N'})$. $N'$ must be large enough, such that the event $\tilde{t}_{N'}$ (corresponding to $\tilde{z}'_N$) happens after $T$. We can easily ensure this by setting $N'$ to some large number (e.g., 100 or 1000), and increasing it if for some sample $\tilde{t}_{N'}$ is less than $T$. As we saw in Figure 7, using larger sequence length leads to no noticeable computational overhead when using GPU.

## C.3   Ensuring that the TPP is valid

We showed in Section 3 that every TPP density $p(\boldsymbol{t})$ corresponds to a differentiable increasing triangular map $\boldsymbol{F}$ defined by the compensator $\Lambda^*$. When directly parametrizing $\boldsymbol{F}$, we need to check one of the two equivalent conditions to ensure that our map $\boldsymbol{F}$ defines a valid temporal point process.

**Condition 1.** The compensator $\Lambda^*(t)$ defined by $\boldsymbol{F}$ must be a continuous function of $t$. (The compensator is already increasing and piecewise-differentiable since $\boldsymbol{F}$ is increasing and differentiable)

**Condition 2.** The map $\boldsymbol{F}$ is bijective (invertible) on the space of increasing sequences. In simple words, we need to ensure that for every increasing sequence $\boldsymbol{z} = (z_1, ..., z_N)$ of arbitrary length $N$, there exists a unique increasing sequence $\boldsymbol{t} = (t_1, ..., t_N)$, such that $\boldsymbol{F}(\boldsymbol{t}) = \boldsymbol{z}$.

## C.4   Parametrizing transformations using splines

Rational quadratic splines used by Durkan et al. [3] define a flexible nonlinear function $g : (0, 1) \to (0, 1)$. When defining our TPP models in Section 3, we need to parametrize functions $\Lambda : [0, T] \to \mathbb{R}_+$ and $\Phi : \mathbb{R}_+ \to \mathbb{R}_+$ that operate on domains different from $(0, 1)$. Moreover, we need to ensure domain compatibility when stacking different transformations, such that the overall transformation $\boldsymbol{F}$ is bijective on the space of increasing sequences (Appendix C.3).

We introduce shortcuts for several helper functions that ensure the domain compatibility

1. $\boldsymbol{\psi}$ applies the function $\psi(x) = 1 - \exp(-x)$ element-wise, where $\psi : \mathbb{R}_+ \to (0, 1)$
2. $\boldsymbol{\psi}^{-1}$ applies the function $\psi^{-1}(y) = -\log(1 - y)$ element-wise, where $\psi^{-1} : (0, 1) \to \mathbb{R}_+$
3. $\boldsymbol{\sigma}$ applies the function $\sigma(x) = 1/(1 + \exp(-x))$ element-wise, where $\sigma : \mathbb{R} \to (0, 1)$
4. $\boldsymbol{\sigma}^{-1}$ applies the function $\sigma^{-1}(p) = \log p - \log(1-p)$ element-wise, where $\sigma^{-1} : (0, 1) \to \mathbb{R}$
5. $\boldsymbol{G}$ applies a rational quadratic spline $g : (0, 1) \to (0, 1)$ element-wise.

We implement the transformation for the modulated renewal process (MRP) as

$$\boldsymbol{F} = \boldsymbol{\psi}^{-1} \circ \boldsymbol{G}_2 \circ \boldsymbol{\psi} \circ \boldsymbol{D} \circ \lambda \boldsymbol{I} \circ \boldsymbol{G}_1 \circ \frac{1}{T}\boldsymbol{I}$$

where $\boldsymbol{I}$ is the identity matrix.

Similarly, we implement the transformation for TriTPP as

$$\boldsymbol{F} = \boldsymbol{\psi}^{-1} \circ \boldsymbol{G}_3 \circ \boldsymbol{\sigma} \circ \boldsymbol{B}_L \circ \cdots \circ \boldsymbol{B}_1 \circ \boldsymbol{\sigma}^{-1} \circ \boldsymbol{G}_2 \circ \boldsymbol{\psi} \circ \boldsymbol{D} \circ \lambda \boldsymbol{I} \circ \boldsymbol{G}_1 \circ \frac{1}{T}\boldsymbol{I}$$

See the code for more details.

## D Datasets

For each synthetic TPP model from Omi et al. [10, Section 4.1], we sampled 1000 sequences on the interval $[0, 100]$. This includes the **Hawkes1**, **Hawkes2**, **self-correcting (SC)**, **inhomogeneous Poisson (IPP)**, **modulated renewal (MRP)** and **renewal (RP)** processes.

**PUBG.**[3] Each sequence contains timestamps of the death of players in a game of Player Unknown's Battleground (PUBG). We use the first 3001 games from the original dataset.

**Reddit-Comments.** Each sequence consists of the timestamps of the comments in a discussion thread posted within 24 hours of the original submission. We consider the submissions to the `/r/askscience` subreddit from 01.01.2018 until 31.12.2019. If several events happen at the *exact* same time, we only keep a single event. The posts are filtered to have a score of at least 100. We collected the data ourselves using the `pushshift` API.[4]

**Reddit-Submissions.** Each sequence contains the timestamps of submissions to the `/r/politics` subreddit within a single day (24 hours). We consider the period from 01.01.2017 until 31.12.2019. If several events happen at the *exact* same time, we only keep a single event. The data is again collected using the `pushshift` API.

**Taxi**[5] contains the records of taxi pick-ups in New York. We restrict our attention to the south of Manhattan, which corresponds to the points with latitude in the interval (40.700084, 40.707697) and longitude in (-74.019871, -73.999443).

**Twitter**[6] contains the timestamps of the tweets by user 25073877, recorded over several years.

**Yelp 1 and 2**[7] contain the user check-in times for the McCarran International Airport and for all businesses in the city of Mississauga in 2018, respectively.

Table 4 shows the number of sequences, average sequence length and the duration of the $[0, T]$ interval for all the datasets.

## E Experimental setup

### E.1 Scalability

For both the RNN-based model and TriTPP we used 20 spline knots. We ran TriTPP with blocks of size $H = 16$ and a total of $L = 4$ block-diagonal layers. This is the configuration of TriTPP with the *largest* number of parameters that we used across our experiments. For the RNN model, we used the hidden size of 32. This is the configuration of the RNN model with the *smallest* number of parameters that we used across our experiments. We did *not* use JIT compilation for either the RNN model or TriTPP, even though enabling JIT would make TriTPP even faster. When measuring the sampling time, we disabled the gradient computation with `torch.no_grad()`. To remove outliers for the RNN model, we removed 10 longest runtimes for both models.

| Dataset name | Number of sequences | Average sequence length | Interval duration |
|---|---|---|---|
| Hawkes 1 | 1000 | 95.4 | 100 |
| Hawkes 2 | 1000 | 97.2 | 100 |
| SC | 1000 | 100.2 | 100 |
| IPP | 1000 | 100.3 | 100 |
| MRP | 1000 | 98.0 | 100 |
| RP | 1000 | 109.2 | 100 |
| PUBG | 3001 | 76.5 | 40 |
| Reddit-C | 1356 | 295.7 | 24 |
| Reddit-S | 1094 | 1129.0 | 24 |
| Taxi | 182 | 98.4 | 24 |
| Twitter | 2019 | 14.9 | 24 |
| Yelp 1 | 319 | 30.5 | 24 |
| Yelp 2 | 319 | 55.2 | 24 |

Table 4: Statistics for the synthetic & real-world datasets

## E.2 Density estimation

**NLL.** In this experiment, we train all models by minimizing the average negative log-likelihood of the training set $\mathcal{D}_{\text{train}} = \{\boldsymbol{t}^{(1)}, \boldsymbol{t}^{(2)}, ...\}$

$$\min_{\boldsymbol{\theta}} -\frac{1}{|\mathcal{D}_{\text{train}}|} \frac{1}{N_{\text{avg}}} \sum_{\boldsymbol{t} \in \mathcal{D}_{\text{train}}} \log p_{\boldsymbol{\theta}}(\boldsymbol{t})$$

We normalize the loss by $N_{\text{avg}}$, the average number of events in a sequence in the training set, in order to obtain values that are at least somewhat comparable across the datasets. We perform full-batch training since the all the considered datasets easily fit into the GPU memory. For all models, we use learning rate scheduling: if the training loss does not improve for 100 iterations, the learning rate is halved. The training is stopped after 5000 epochs or if the validation loss stops improving for 300 epochs, whichever happens first. We train all models using the parameter configurations reported in Section 6.2 and pick the configuration with the best validation loss.

**MMD.** We train the models & tune the hyperparameters using the same procedure as in the NLL experiment. Then, we compare the distribution $p(\boldsymbol{t})$ learned by each model with the empirical distribution $p^{\star}(\boldsymbol{t})$ on the hold-out test set by estimating the maximum mean discrepancy (MMD) [59]. The MMD between distributions $p$ and $p^{\star}$ is defined as

$$\text{MMD}(p, p^{\star}) = \mathbb{E}_{\boldsymbol{t}, \boldsymbol{t}' \sim p}[k(\boldsymbol{t}, \boldsymbol{t}')] - 2\mathbb{E}_{\boldsymbol{t} \sim p, \boldsymbol{u} \sim p^{\star}}[k(\boldsymbol{t}, \boldsymbol{u})] + \mathbb{E}_{\boldsymbol{u}, \boldsymbol{u}' \sim p^{\star}}[k(\boldsymbol{u}, \boldsymbol{u}')]$$

Here, $\boldsymbol{t} = (t_1, ..., t_N)$ and $\boldsymbol{u} = (u_1, ..., u_M)$ denote variable-length TPP realizations from different distributions, and $k(\cdot, \cdot)$ is a positive semi-definite kernel function that quantifies the similarity between two TPP realizations. We use the Gaussian kernel

$$k(\boldsymbol{t}, \boldsymbol{u}) = \exp\left(-\frac{d(\boldsymbol{t}, \boldsymbol{u})}{2\sigma^2}\right)$$

where $d(\boldsymbol{t}, \boldsymbol{u})$ is the counting measure distance between two TPP realizations from [60, Equation 3], defined as

$$d(\boldsymbol{t}, \boldsymbol{u}) = \sum_{i=1}^{N} |t_i - u_i| + \sum_{i=N+1}^{M} (T - u_i)$$

Figure 10: The blue area represents the counting measure distance (figure adapted from [60]).

Here, we assume w.l.o.g. that $N \leq M$. Following Section 8 of Gretton et al. [59], the parameter $\sigma$ is estimated as the median of $d(\boldsymbol{t}, \boldsymbol{u})$ with $\boldsymbol{t}, \boldsymbol{u} \sim p \cup p^{\star}$.

### E.3 Variational inference

We simulate an MMPP with $K = 3$ states and the following parameters

$$\boldsymbol{A} = \begin{pmatrix} 0.1 & 0.1 & 0.1 \\ 0.1 & 0.1 & 0.1 \\ 0.1 & 0.1 & 0.1 \end{pmatrix} \qquad \boldsymbol{\pi} = \begin{pmatrix} 0.52 \\ 0.22 \\ 0.26 \end{pmatrix} \qquad \boldsymbol{\lambda} = \begin{pmatrix} 1 \\ 5 \\ 20 \end{pmatrix}$$

We use the following configuration for TriTPP in this experiment: $L = 2$ blocks of size $H = 4$, learning rate 0.01, no weight decay. We estimate the ELBO using 512 Monte Carlo samples from $q(\boldsymbol{t})$ and use the temperature $\gamma = 0.1$ for the relaxation. We implemented the MCMC sampler by Rao & Teh [49] in Pytorch. We discard the first 100 samples (burn-in stage), and use 1000 samples to compute the marginal distribution of the posterior.

## F  Additional experiments

### F.1  Density estimation

**NLL table with standard deviations.** For most models & datasets the results are nearly independent of the random initialization and the standard deviations are very close to zero. In the following table, we show the standard deviations of the NLL computed over 5 random initializations for all datasets where at least one of the models has the standard deviation above 0.005.

|  | PUBG | | Reddit-S | | Twitter | | Yelp2 | |
|---|---|---|---|---|---|---|---|---|
|  | mean | std | mean | std | mean | std | mean | std |
| TriTPP | -2.41 | 0.34 | -4.49 | 0.06 | 1.06 | 0.01 | -0.09 | 0.01 |
| RNN | -1.97 | 0.16 | -4.89 | 0.29 | 1.08 | 0.01 | -0.07 | — |
| MRP | -0.83 | 0.08 | -4.38 | 0.05 | 1.23 | — | -0.1 | — |
| RP | 0.12 | — | -4.01 | 0.03 | 1.2 | — | -0.02 | — |
| IPP | -0.06 | — | -4.08 | — | 1.61 | — | -0.05 | — |

Table 5: Average test set NLL with standard deviations. Datasets where all models have a standard deviation below 0.005 are excluded.

**Effect of the block size and number on TriTPP performance.** In this experiment, we show that TriTPP works well with different numbers $L$ and sizes $H$ of block-diagonal layers. We use the same setup as in the density estimation experiment. Table 6 shows the test set NLL scores for different configurations. Smaller block are helpful for datasets with a clear global trend (e.g., Reddit-S, Taxi, Yelp), and larger blocks help for datasets with bursty behavior (Reddit-C, Twitter). In all cases, TriTPP is better than simpler baselines, like MRP, RP and IPP (Table 1).

| Configuration | Hawkes1 | Hawkes2 | SC | IPP | MRP | RP | PUBG | Reddit-C | Reddit-S | Taxi | Twitter | Yelp1 | Yelp2 |
|---|---|---|---|---|---|---|---|---|---|---|---|---|---|
| TriTPP ($L = 2, H = 4$) | 0.58 | 0.01 | 0.86 | 0.71 | 0.35 | 0.24 | -0.95 | -2.26 | -4.69 | -0.68 | 1.11 | 0.62 | -0.1 |
| TriTPP ($L = 4, H = 4$) | 0.57 | 0.01 | 0.85 | 0.71 | 0.35 | 0.24 | -2.04 | -2.28 | -4.57 | -0.68 | 1.06 | 0.63 | -0.1 |
| TriTPP ($L = 2, H = 8$) | 0.56 | 0.01 | 0.84 | 0.71 | 0.35 | 0.24 | -1.93 | -2.3 | -4.42 | -0.66 | 1.06 | 0.64 | -0.09 |
| TriTPP ($L = 4, H = 8$) | 0.56 | 0.0 | 0.83 | 0.71 | 0.35 | 0.24 | -2.41 | -2.33 | -4.46 | -0.67 | 1.06 | 0.64 | -0.09 |
| TriTPP ($L = 2, H = 16$) | 0.56 | 0.0 | 0.84 | 0.71 | 0.36 | 0.25 | -1.78 | -2.35 | -4.45 | -0.64 | 1.06 | 0.67 | -0.06 |
| TriTPP ($L = 4, H = 16$) | 0.56 | 0.0 | 0.84 | 0.72 | 0.36 | 0.25 | -1.83 | -2.36 | -4.49 | -0.64 | 1.07 | 0.67 | -0.06 |

Table 6: Test set NLL for different configurations of TriTPP.

**Visualizing the effect block-diagonal matrices.** A completely arbitrary compensator $\Lambda^*$ leads to a completely arbitrary increasing triangular map $\boldsymbol{F}$. However, by picking a parametric class of models, such as MRP or TriTPP, we restrict the set of possible maps $\boldsymbol{F}$ that our model represent. One way to visualize the dependencies captured by the map $\boldsymbol{F}$ is by looking at its Jacobian $J_{\boldsymbol{F}}$.

Figures 11 and 12 show the Jacobians of the component transformations for the modulated renewal process and TriTPP. We can obtain the overall (accumulated) Jacobian of the entire transformation by multiplying the component Jacobians from right to left. We can see that thanks to the block-diagonal layers TriTPP is able to capture more complex transformations, and thus richer densities, than MRP.

Figure 11: Jacobians of the component transformations of the modulated renewal process. We obtain the Jacobian of the combined transformation $\boldsymbol{F} = \boldsymbol{C} \circ \boldsymbol{\Phi} \circ \boldsymbol{D} \circ \boldsymbol{\Lambda}$ by multiplying the Jacobians of each transform (right to left).

Figure 12: Jacobians of the component transformations of TriTPP. We obtain the Jacobian of the combined transformation $\boldsymbol{F} = \boldsymbol{C} \circ \boldsymbol{\Phi}_2 \circ \boldsymbol{B}_4 \circ \boldsymbol{B}_3 \circ \boldsymbol{B}_2 \circ \boldsymbol{B}_1 \circ \boldsymbol{\Phi}_1 \circ \boldsymbol{D} \circ \boldsymbol{\Lambda}$ by multiplying the Jacobians of each transform (right to left).

**Distribution of sequence lengths.** In this experiment, we additionally quantify how well each model captures the true data distribution. Like before, we train all models on the training set. We then generate sequences $\boldsymbol{t}$ from a trained model and compare the distribution of their lengths to the distribution of the lengths of the true data using Wasserstein distance. We use the whole dataset since the test sets is too small in some cases. Using Python pseudocode, this procedure can be expressed as

```
lengths_sampled = [len(t) for t in model_samples]
lengths_true = [len(t) for t in dataset]
wd = wasserstein_distance(lengths_sampled, lengths_true)
```

Figure 13 shows the distributions for the Twitter dataset together with the respective Wasserstein distances. Note that the histograms are used only for visualization purposes, the Wasserstein distance is computed on the raw distributions. Quantitative results are reported in Table 7. We observe the same trend as before: the RNN-based model and TriTPP consistently outperform the other methods. Recall that Hawkes process achieves a good NLL on the Twitter data (Table 1). However, when we sample sequences from the trained Hawkes model, the distribution of their lengths doesn't actually match the true data, as can be seen in Figure 13c.

(a) TriTPP (WD = 0.17)          (b) RNN (WD = 0.10)          (c) Hawkes (WD = 0.50)

Figure 13: Histograms of sequence lengths (true and generated) for Twitter. The difference between the two is quantified using Wasserstein distance (WD) — lower is better.

| | Hawkes1 | Hawkes2 | SC | IPP | MRP | RP | PUBG | Reddit-C | Reddit-S | Taxi | Twitter | Yelp1 | Yelp2 |
|---|---|---|---|---|---|---|---|---|---|---|---|---|---|
| IPP | 0.11 | 0.11 | 0.03 | **0.00** | 0.03 | 0.07 | **0.01** | 0.76 | 0.27 | 0.10 | 0.52 | 0.07 | 0.12 |
| RP | 0.07 | 0.09 | 0.19 | 0.14 | 0.38 | 0.02 | 0.67 | 0.67 | 0.28 | 0.85 | 0.28 | 0.31 | 0.20 |
| MRP | 0.08 | 0.07 | 0.02 | **0.00** | **0.01** | 0.01 | 0.05 | 0.66 | 0.27 | 0.09 | 0.28 | 0.06 | 0.11 |
| Hawkes | 0.01 | 0.05 | 0.03 | 0.15 | 0.15 | 0.03 | 0.12 | **0.25** | 0.65 | 0.09 | 0.50 | 0.10 | 0.15 |
| RNN | **0.00** | **0.01** | **0.00** | 0.04 | 0.03 | **0.00** | 0.08 | 0.40 | **0.07** | **0.08** | **0.10** | **0.05** | 0.12 |
| TriTPP | 0.05 | 0.03 | **0.00** | 0.01 | **0.01** | **0.00** | 0.05 | 0.53 | 0.24 | **0.08** | 0.17 | **0.05** | **0.09** |

Table 7: Wasserstein distance between the distributions of lengths of true and sampled sequences.

## F.2 Variational inference

**Random initliazations.** In order to show that ours results are not cherry-picked, we provide the plots of marginal posterior trajectories (similar to Figure 8) obtained with 3 different random seeds. Figure 14 shows that our results are consistent across the random seeds.

Figure 14: Marginal posterior trajectories obtained when using different random seeds.

**Variational inference on real-world data.** We apply our model to the server log data [8]. More specifically, we perform segmentation on the interval that contains the first 200 events. We estimate the posterior over the trajectories $(t, s)$ and learn the model parameters $\theta = \{\pi, A, \lambda\}$ using the procedure described in Equation 16. Like before, we compare our approach to the MCMC sampler of Rao & Teh. For the MCMC sampler, we adopt an EM-like approach, where we alternate between closed-form parameter updates for $\theta$ and simulating the posterior trajectories. Figure 15 shows the obtained posterior trajectories for the two approaches. Both models learn to segment the sequence into a high-event-rate and a low-event-rate states.

Figure 15: Segmentation of server data obtained using our VI approach and MCMC. In both cases, we estimate the posterior $p(t, s | o, \theta)$ as well as the MMPP parameters $\theta$.

### F.3 Miscellaneous

**Convergence plots for density estimation.**

(a) Twitter — TriTPP

(b) Taxi — TriTPP

(c) PUBG — TriTPP

(d) Twitter — RNN

(e) Taxi — RNN

(f) PUBG — RNN

Figure 16: Training loss convergence for TriTPP and RNN model with different random seeds.

**Convergence plots for variational inference.**

Figure 17: Convergence of our variational inference procedure when using 5 different random seeds.