[Reviews · NeurIPS 2020]

Review 1

Summary and Contributions: In this paper, the authors propose a temporal point process model that allows efficient sampling and likelihood computation. The proposed model is applied to minimized sample-based loss functions. The authors apply the model to density estimation tasks and MMPP inference problems. **post-rebuttal**: I have read the authors' rebuttal and would like to keep my rating.

Strengths: The paper is well-motivated by noting the connection between the likelihood functions between a TPP and a triangular map. The authors also elaborate on ways to speed up the proposed model to leverage modern computational devices. The proposed model can be applied to various sample-based loss functions.

Weaknesses: The experimental results seem weak. Although there's computational advantages, the proposed model seems to underperform the existing and simple baseline models on most datasets.

Correctness: I believe the technical aspect of the paper is solid and experiments are properly designed.

Clarity: The paper is well organized and clearly written. I found it relatively easy to follow.

Relation to Prior Work: To the best of my knowledge, the authors present a thorough overview of the existing work, and clearly distinguish this work from them.

Reproducibility: Yes

Additional Feedback:


Review 2

Summary and Contributions: this is excellent work that makes a connection between normalizing flows and temporal point processes. it was a pleasure to read. empirically, the authors show that performance is comparable to recurrent neural networks, while prediction and training are much faster.

Strengths: the claims are sound. the connection between temporal point processes and normalizing flows, while straightforward, are novel and significant. i anticipate several extensions to this work that will further improve its empirical performance. the authors do an excellent empirical evaluation of the claims to compare triangular map-based temporal point processes to state of the art methods.

Weaknesses: the performance of triangular temporal point processes is similar to recurrent neural network models. however, i anticipate that further improvements in normalizing flows can be translated to improved triangular map-based temporal point processes. for example, i am not sure how the authors chose the block size relative to the RNN hidden state size (were these set to match the number of parameters?). similarly, incorporating additional nonlinearities may improve performance.

Correctness: yes. the authors do an excellent job of describing the experimental set up, with hyperparameter settings and links to related work. it seems straightforward to reproduce the results.

Clarity: yes. this was the best-written paper i reviewed in this cycle.

Relation to Prior Work: yes.

Reproducibility: Yes

Additional Feedback: it is hard to build intuition on triangular temporal point processes. maybe clearly describing how the pairwise difference matrix compares to something readers are already familiar with (such as a recurrent neural network architecture). similarly, i think you can improve the empirical performance with stochastic gradient descent instead of an adaptive learning rate optimizer, at the cost of careful tuning of momentum and gradient clipping. (related to the above point, i am not sure how to think of gradient clipping in triangular temporal point processes -- gradients will explode in accord with the singular values of the block matrices in the compisition?) having a real-world example will also help readers. right now reddit comments are maybe the most real-world scenario. bringing this up earlier or choosing another running example could help understand the work if people are less familiar with temporal point processes like myself. (similarly, giving a real-world example of markov jump processes, such as from neuroscience, could also help.) small typo: "we implemented tritpp and rnn model using pytorch" nits: personifying TriTPP: 'the key feature of tritpp is its ability', and RNNs are not personified. better to be consistent.


Review 3

Summary and Contributions: The paper showed there is a one to one correspondence between TPP densities and increasing triangular maps. The TPP models can be constructed directly by specifying the respective transformations, for which both density evaluation and sampling can be done efficiently in parallel.

Strengths: The theory and experiments are solid. It is interesting to find the correspondence between TPP densities and increasing triangular maps. By parametrizing the transformation F tactfully, the sampling procedure can be acclerated which enables the sampling-based inference for TPP models.

Weaknesses: One concern is about the sampling procedure. The sampling procedure is first drawing z from an unit rate HPP and secondly applying the inverse map to get t=F^{-1}(z). If my understanding is right, although the evaluation of F^{-1} is parallelized, the sampling from an unit rate HPP is still sequential. So, in essence, the algorithm is not completely parallel.

Correctness: It is correct.

Clarity: It is well written. Typo: line 331 Fig6--> Fig8

Relation to Prior Work: The paper discussed the difference with previous contributions in normalzing flow, different models and sampling methods in point processes.

Reproducibility: Yes

Additional Feedback: UPDATE: I have read and taken into account the rebuttal and my opinion has not changed.


Review 4

Summary and Contributions: This work first proposes a new parametrization for several classic temporal point processes (TPPs), which enables efficient parallel likelihood computation and sampling. TPP allows to naturally handle data that consists of variable-number events in continuous time. These classic TTP models with existing parametrization was inherently sequential. Next, the authors proposed a new class of non-recurrent TPP models, namely TriTPP, where both sampling and likelihood computation can be done in parallel. TPP models combined with recurrent neural networks provide a highly flexible powerful framework, but still remain sequential, making TPPs poorly suited for sampling. The proposed TriTPP matches the flexibility of RNN-based methods, while allowing orders of magnitude faster sampling. Finally, the authors derive a differentiable relaxation for non-differentiable sampling-based TPP losses, which allows a new variational inference scheme for Markov jump processes. The first result arrives from identifying the fact that TPP densities can be represented by increasing triangular maps. The second result is achieved by adding multiple learnable block-diagonal matrices, with each block being a repeated lower-triangular matrix with strictly positive diagonal entries, into the decomposition of the triangular map F. **post-rebuttal**: I have read the authors' rebuttal and would like to keep my rating.

Strengths: The paper was well written and presented, making it easy to follow. The supplementary material includes codes for reproducing the results. The major strength of the proposed model TriTPP is scalability, i.e., efficient parallel sampling, while maintaining the flexibility of the RNN based TPP model, which means the proposed method can accurately learn the data distribution. The reported experimental results on synthetic and real data sets support these claims. The paper also shows how the flexibility and efficiency of TriTPP allow it to be used as a plug-and-play component in other machine learning models.

Weaknesses: I did not notice any.

Correctness: The main claims are two folds: 1) TriTPP can perform likelihood computation and sampling in parallel, which is true from the decomposition of the triangular map F in line 139. The matrices, their derivatives, and inverses are computable in closed form to meet the efficiency requirements. 2) TriTPP can maintain the flexibility of the RNN based TPP model. This claim is mainly supported by experimental results on synthetic and real data sets in Section 6.2. This flexibility in TriTPP seems to come from the block diagonal matrices and from shifting the blocks in every other layer to capture long-range dependencies.

Clarity: Yes, the paper is well written and organized.

Relation to Prior Work: Yes, this has been addressed in sections 3.2 and 5.

Reproducibility: Yes

Additional Feedback:

[Author Response · NeurIPS 2020]

## 1 All reviewers

We would like to thank all the reviewers for their positive assessment of our work. We are also grateful for pointing out the typos and providing writing suggestions — we will incorporate them in the updated version.

## 2 Reviewer 1

- **Experimental results.**

    We would like to politely disagree with your statement that "the proposed model seems to underperform the existing and simple baseline models on most datasets". In the density estimation experiments (Tables 1 & 2), our method achieves the best or the second best score in 19/26 cases. RNN is the only model besides TriTPP that consistently achieves good results across all the datasets. In the NLL experiment TriTPP dominates the simpler baselines (IP, RP, MRP models) on Hawkes1, Hawkes2, SC, PUBG, Reddit-C, Reddit-S, Twitter (7/13), and achieves nearly identical scores on the remaining datasets (6/13). Similarly, we convincingly outperform the Hawkes model on Hawkes2, SC, IPP, MRP, RP, PUBG, Reddit-S, Yelp1, Yelp2 datasets (9/13), get slightly better scores on Taxi and Twitter (2/13) and get worse scores only on the Hawkes1 and Reddit-C datasets (2/13).

    We would also like to reiterate that the parallelism of our model is benefecial not only for faster training on long sequences. Rather, fast parallel sampling opens new applications for TPPs that were completely off the charts for existing approaches (where sampling can be hundreds of times slower).

## 18 Reviewer 2

- **Choice of the block size.**

    We found that increasing the block size beyond 16 for the TriTPP model did not improve the performance. In contrast, the RNN did benefit from larger hidden sizes (32 or 64). Therefore, we allowed the RNN to have more parameters in the density estimation experiment (Section 6.2), not to give an unfair advantage to TripTPP. In the scalability experiment (Section 6.1) we made sure that both models have approximately the same number of parameters.

- **Additional nonlinearities.**

    We found that stacking the nonlinearities (splines) did not improve the performance on the validation set. One explanation for this can be that the splines do not benefit from depth, as their flexibility can be increased by simply increasing the number of knots.

- **Using SGD & tuning the optimization procedure.**

    Thank you, we will run more experiments to see if this will allow us to improve the performance.

- **More examples & intuition for TPPs and triangular maps.**

    Thank you for the excellent suggestion — we will add more examples as well as the following discussion to help the readers build intuition on this topic.

    The function $\Lambda$ captures the global trend by rescaling the time (e.g., if there are more events during the weekend than on a weekday). The pairwise difference matrix $D$ computes the "rescaled" inter-event times after the global trend component $\Lambda$ has been removed. The block-diagonal matrices $B_1, ..., B_L$ capture the interactions between events, which allows our model to capture self-exciting or self-correcting behavior. The function $\Phi$ models the distribution of the "raw" inter-event times (this may correspond to a recurring pattern, as in a renewal process). Finally, the cumulative sum matrix $C$ sums up the transformed inter-event times to obtain the arrival times of the base unit Poisson process.

## 41 Reviewer 3

- **Sampling from a Poisson process.**

    We would like to point out that we can sample from a unit-rate homogeneous Poisson process in parallel as following (using Pytorch as an example):

    ```
    tau = torch.empty(N).exponential_(1.0)    # draw inter-event times from Expo(1)
    t = tau.cumsum(dim=-1)                     # obtain HPP arrival times
    ```

    Both steps can be done in parallel in $O(N)$ operations, including the cumulative sum (Blelloch, 1990. "Prefix sums and their applications").

[Meta-Review · NeurIPS 2020]

This work draws a connection between temporal point processes and normalizing flows/triangular maps. While there were concerns about the computational experiments, reviewers agreed that this is novel and interesting work that is likely to inspire new research.